PREPARED FOR SUBMISSION TO SCIPOST PHYSICS

DESY-25-066

# Sparsity in the numerical six-point bootstrap

**Sebastian Harris**[a,b]

[a] *Deutsches Elektronen-Synchrotron DESY, Notkestr. 85, 22607 Hamburg, Germany*
[b] *Zentrum für Mathematische Physik, Bundesstrasse 55, 20146 Hamburg, Germany*

*E-mail:* sebastian.harris@desy.de

ABSTRACT: The paper contributes to an ongoing effort to extend the conformal bootstrap beyond its traditional focus on systems of four-point correlation functions. Recently, it was demonstrated that semidefinite programming can be used to formulate a six-point generalisation of the numerical bootstrap, yielding qualitatively new, rigorous bounds on CFT data. However, the numerical six-point bootstrap requires solving SDPs involving infinite-dimensional matrices, which has so far limited its applicability and hindered scalability in early implementations. This work overcomes the challenges by using sparse matrix decompositions to exploit the banded structure of the underlying SDP. The result is a rewriting of one-dimensional six-point bootstrap problems as effectively two-dimensional standard mixed correlator four-point bootstrap computations. As application, novel bounds whose extremal correlators interpolate between the six-point functions of the generalised free fermion and boson are derived. The extremal interpolations are matched with perturbative deformations of the massive free boson in $AdS_2$.

# 1 Introduction

Over the last two decades[1], the numerical conformal bootstrap [3] has become an essential asset in the toolbox of the modern theoretical physicist. Like most great tools, it is a versatile instrument that serves its user in various ways.

From a quantitative perspective, the bootstrap offers access to scaling dimensions and OPE coefficients of CFTs as one tool among many, such as, if applicable, Monte Carlo simulations, Hamiltonian truncation, fuzzy sphere regulation, perturbation theory and others. For many theories it outperforms all available alternatives. Famously this is the case, for example, in the 3d Ising model [4–10] as well as in more general small $N$ bosonic $O(N)$-fixed points [8, 11–14] and $O(N)$-symmetric Gross-Neveu-Yukawa fixed points [15–17].

From a qualitative perspective, the bootstrap is unique. It allows us to answer questions about the space of all CFTs in a theory agnostic fashion. This makes it a highly valuable exploratory tool, challenging and inspiring us to think about CFTs in new ways. The purpose of this paper is to contribute to an ongoing effort to enlarge the type of questions that we can answer through the numerical bootstrap.

Answering new questions typically requires to make new sources of information accessible. In the context of this paper, the new source of information is the set of crossing equations of six-point correlation functions. Studying the crossing equations of higher-point correlators has been fruitful in the analytic bootstrap and, for instance, led to various results on large spin asymptotics of OPE coefficients and multi-twist operators in lightcone bootstrap works such as [18–22]. The first applications of numerical techniques to the investigation of higher-point correlators were [23] and its follow ups [24, 25]. In those two works, numerical solutions to truncated crossing equations of five-point functions were determined in order to obtain approximate conformal data of the 3d Ising CFT. As usual with truncation based approaches to crossing, this five-point bootstrap involved unknown systematic errors that have to carefully be estimated through suitable heuristics. A six-point bootstrap developed in the same year [26] demonstrated that SemiDefinite Programming (SDP) can be used to instead generate exact bounds from higher-point correlators.

Through the crossing relations of six-point functions, one can naturally answer questions on triple OPEs just as the bootstrap analysis of four-point functions allows us to answer questions on the fusion of pairs of operators. Let us make this more precise by considering the example of correlation functions of a scalar field $\phi$ in an arbitrary (unitary) CFT. The OPE of two copies of $\phi$ takes the form

$$\phi(x_1)\phi(x_2) \sim \sum_{\mathcal{O} \in \phi \times \phi} F_{\phi\phi\mathcal{O}}(x_1, x_2)\mathcal{O}(x_2), \tag{1.1}$$

where the function $F_{\phi\phi\mathcal{O}}(x_1, x_2)$ is kinematically fixed up to an overall normalisation constant, namely the OPE coefficient $C_{\phi\phi\mathcal{O}}$. By studying crossing of the $\langle\phi\phi\phi\phi\rangle$ four-point function, the numerical bootstrap can answer a large range of questions on the fields $\mathcal{O} \in \phi \times \phi$, putting bounds on the scaling dimensions $\Delta_{\mathcal{O}}$ and the OPE coefficients $C_{\phi\phi\mathcal{O}}$.

---

[1]See [1] for a review of the latest developments and [2] for earlier results.

Consider now the triple OPE

$$\phi(x_1)\phi(x_2)\phi(x_3) \sim \sum_{\mathcal{O} \in \phi \times \phi \times \phi} F_{\phi\phi\phi\mathcal{O}}(x_1, x_2, x_3)\mathcal{O}(x_3). \qquad (1.2)$$

In this case, conformal symmetry of course still constrains $F_{\phi\phi\phi\mathcal{O}}(x_1, x_2, x_3)$. However, the kinematic constraints are not strong enough to completely fix the functional form of $F_{\phi\phi\phi\mathcal{O}}$. Just as the crossing symmetry of the four-point function is a dynamical input that allows the numerical bootstrap to answer questions about the $\phi \times \phi$ OPE, crossing equations of the six-point function $\langle\phi\phi\phi\phi\phi\phi\rangle$ enable the bootstrap to answer questions on the $\phi \times \phi \times \phi$ triple OPE, putting bounds on the dimensions $\Delta_{\mathcal{O}}$ of the operators $\mathcal{O} \in \phi \times \phi \times \phi$ as well as on the values of the function $F_{\phi\phi\phi\mathcal{O}}$ and its derivatives [2].

Reference [26] has demonstrated that the idea described in the previous paragraph can indeed be implemented in practice. Its authors have for the first time successfully extracted rigorous bounds on triple OPE CFT data by analysing six-point crossing equations with SDP techniques. However, the initial implementation suffered from various technical difficulties that limited its range of applicability. On the one hand, it had to resort to a discretisation of the spectrum of exchanged operators and could not rely on polynomial SDP techniques that have been the standard in four-point bootstrap since [7]. On the other hand, SDPB – the highly optimised solver designed specifically to address the needs of the bootstrap community – is not well suited to solve the SDPs encountered in [26], which forced the authors to resort to using the general purpose solver SDPA [27]. Support for the latter has however seized ten years ago and, crucially, no arbitrary precision, parallelisable version of SDPA is available. This fact is a major bottle neck for SDPA based approaches to bootstrap, making it virtually impossible to scale programs up and benefit from the computational power of modern clusters.

The present paper reports on a conceptual improvement of the six-point bootstrap which allows us to overcome the two mayor obstacles that have been described in the previous paragraph. Using sparse SDP techniques, the banded structure of the infinite matrices that occur in the six-point bootstrap of [26] can be efficiently exploited. The final result is that 1d six-point bootstrap problems reduce to SDPs that are of the same type as those encountered in finite mixed correlator systems of 2d four-point bootstrap. Since SDPB was specifically designed to excel at solving SDPs of that type, it can thus directly be applied to the six-point bootstrap. This both removes the need of discretisations and makes large scale parallelisation applicable. These technical improvements extend the possibilities for six-point bootstrap applications, which is demonstrated through two examples in this paper.

As a first example, Section 4.1 revisits the "gap maximisation without identity" problem that has been studied in [26]. In the previous formulation of the six-point bootstrap it was

---

[2]Note that, in principle, all data encoded in the triple OPE should also be accessible through four-point functions. More concretely, it is conceivable that the bounds that we can obtain through the six-point bootstrap could in some way also be determined by studying a large mixed correlator system of four-point functions involving correlation functions of the type $\langle\mathcal{O}\phi\phi\mathcal{O}'\rangle$ with $\mathcal{O}, \mathcal{O}' \in \phi \times \phi$. However, it seems unclear how such an alternative approach would be implemented in practice.

even challenging to obtain the 15 functional (four derivative) bounds illustrated in Figure 7 of [26]. With the refined method presented in the current paper, the bound was determined with up to 53 functionals (seven derivatives), while at the same time also scanning over a much finer grid of external scaling dimensions, see Figure 2.

The main application of numerical six-point bootstrap studied in this paper is described in Sections 4.2, 4.3 and 4.4. Concretely, a novel two parameter family of extremal solutions to crossing interpolating between the Generalised Free Boson (GFB) and Generalised Free Fermion (GFF) is found using the new numerical tools. A similar, OPE maximising, interpolation between GFB and GFF has been studied before, using the four-point bootstrap, for instance in [28]. Close to GFB, the four-point extremal solution was found to match the $\lambda_4\Phi^4$ deformation of the massive free boson in AdS$_2$. Likewise, we show that, around the GFB point, the plane tangent to the interpolating surface obtained from the six-point bootstrap is spanned by the $\lambda_4\Phi^4$ and $\lambda_6\Phi^6$ deformation.

**Outline.** First, Section 2 provides a concise review of the six-point crossing equations, derivative functionals for them and the numerical six-point bootstrap. Section 3 then describes in detail the main technical improvements of this paper. After the review Section 3.1, which discusses sparse SDP for banded matrices in general terms, Section 3.2 specialises to the particular SDPs that arise in the numerical six-point bootstrap. As already outlined in the previous paragraphs, Section 4 describes two applications of the improved six-point bootstrap. The paper ends in Section 5 with a discussion of potential future research directions. The ancillary files available on the arXiv page of this paper should allow the reader to reproduce the bounds presented in Section 4. Appendix A contains instructions on how to use the corresponding code.

## 2 Semidefinite programming and six-point crossing

This section briefly reviews the semidefinite programming based numerical six-point bootstrap of Reference [26]. It starts in Section 2.1 with a discussion of the crossing equation, which is followed by a description of derivative functionals that act on that equation in Section 2.2. Finally, the dual SDP formulation of the six-point crossing equation and some aspects of its numerical implementation are discussed in Section 2.3.

### 2.1 The crossing equation

Throughout this paper, we study conformally invariant six-point functions of identical scalars $\phi_i := \phi(x_i)$ inserted on points $x_i < x_{i+1}$ on the line. Such a correlator can always be reduced to a function of three cross-ratios $\chi_i$ by extracting a suitable kinematic prefactor $\mathcal{L}(x_i)$. More concretely, let us use $x_{ij} := x_i - x_j$ and $x_{i+6} := x_i$ to define

$$\chi_i := \frac{x_{i,i+1}\,x_{i+2,i+3}}{x_{i,i+2}\,x_{i+1,i+3}} \qquad \text{and} \qquad \mathcal{L}(x_i) := (x_{12}^2 x_{34}^2 x_{56}^2 \chi_2)^{-\Delta_\phi} \qquad (2.1)$$

such that the reduction of the correlator to a function $G$ of cross-ratios becomes

$$\langle \phi_6\phi_5\phi_4\phi_3\phi_2\phi_1 \rangle = \mathcal{L}(x_i)G(\chi_1, \chi_2, \chi_3). \qquad (2.2)$$

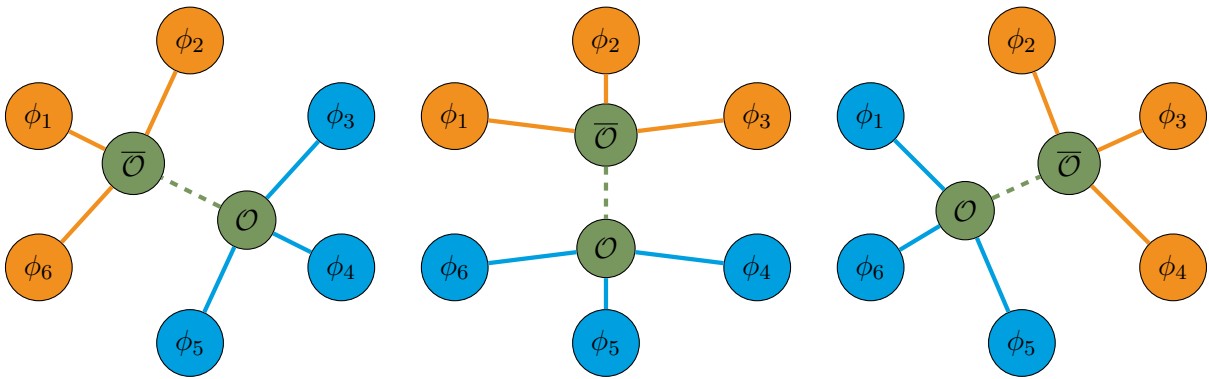

**Figure 1**: The three different decompositions of a six-point correlator into products of four-point functions.

By inserting projectors onto conformal multiplets, the six-point function can be written as a sum of products of lower-point correlators. In particular, reducing it to products of the kinematically fixed three-point functions results in the conformal block decomposition of the correlator. For our purposes, the relevant decomposition is that into four-point functions. As usual in the conformal bootstrap, we leverage the fact that there are multiple inequivalent decompositions of the same correlator, leading to a power full set of functional equations – the crossing equations. The three different possibilities for decomposing the correlator (2.2) into products of four-point functions are illustrated in Figure 1.

In equations, the central decomposition in Figure 1 is

$$\langle \phi_6 \phi_5 \phi_4 \phi_3 \phi_2 \phi_1 \rangle = \sum_{\mathcal{O}} \sum_{n=0}^{\infty} \frac{\langle \phi_6 \phi_5 \phi_4 P^n \mathcal{O} \rangle \langle \mathcal{O}^\dagger K^n \phi_3 \phi_2 \phi_1 \rangle}{(2\Delta_{\mathcal{O}})_n n!}, \tag{2.3}$$

which was shown in Appendix A of [26] to reduce to

$$G(\chi_1, \chi_2, \chi_3) = \sum_{\mathcal{O}} \sum_{n=0}^{\infty} F_n^{\mathcal{O}}(\chi_1) \frac{\chi_2^{\Delta_{\mathcal{O}}+n}}{(2\Delta_{\mathcal{O}})_n n!} F_n^{\mathcal{O}}(\chi_3), \tag{2.4}$$

where, for $0 < \chi < 1$,

$$F_n^{\mathcal{O}}(\chi) := (-1)^n (\bar{\chi})^{\Delta_{\mathcal{O}}+n-\Delta_\phi} \partial_{\bar{\chi}}^n f^{\mathcal{O}}(\bar{\chi}) \tag{2.5}$$

is defined in terms of

$$\bar{\chi} := \chi^{-1} \qquad \text{and} \qquad f^{\mathcal{O}}(\bar{\chi}) := \langle \phi(\infty) \mathcal{O}(\bar{\chi}) \phi(1) \phi(0) \rangle. \tag{2.6}$$

The other decompositions can be obtained by shifting $x_i \to x_{i\pm1}$. Therefore, the functions $F_n^{\mathcal{O}}$ obey the functional equation[3]

---

[3]Note that $\chi_4$ and the ratio $\frac{\mathcal{L}(x_{i+1})}{\mathcal{L}(x_i)}$ can be expressed in terms of $\chi_1$, $\chi_2$ and $\chi_3$ as

$$\chi_4 = \frac{\chi_1 + \chi_2 + \chi_3 - \chi_1\chi_3 - 1}{\chi_1 + \chi_2 - 1} \qquad \text{and} \qquad \frac{\mathcal{L}(x_{i+1})}{\mathcal{L}(x_i)} = \left( \frac{\chi_1^2 \chi_2 \chi_3}{(\chi_1 + \chi_2 + \chi_3 - \chi_1\chi_3 - 1)^2} \right)^{\Delta_\phi}. \tag{2.7}$$

Thus, eq. (2.8) is a functional equation depending on the three cross-ratios only.

$$\sum_{\mathcal{O}} \sum_{n=0}^{\infty} \left[ F_n^{\mathcal{O}}(\chi_1) \frac{\chi_2^{\Delta_{\mathcal{O}}+n}}{(2\Delta_{\mathcal{O}})_n n!} F_n^{\mathcal{O}}(\chi_3) - \frac{\mathcal{L}(x_{i+1})}{\mathcal{L}(x_i)} F_n^{\mathcal{O}}(\chi_2) \frac{\chi_3^{\Delta_{\mathcal{O}}+n}}{(2\Delta_{\mathcal{O}})_n n!} F_n^{\mathcal{O}}(\chi_4) \right] = 0 \qquad (2.8)$$

whose solutions are bootstrapped in this paper.

## 2.2 The functionals

As usual in the bootstrap, we need to act with certain functionals on the crossing equation (2.8) to extract equations that do not depend on undetermined cross-ratios anymore. Concretely, we act with combinations $\partial_{\chi_1}^a \partial_{\chi_2}^b \partial_{\chi_3}^c$ of derivatives on eq. (2.8).

Before we do so observe that the functions $F_n^{\mathcal{O}}$ defined in eq. (2.5) satisfy

$$\chi \partial_\chi F_n^{\mathcal{O}}(\chi) = F_{n+1}^{\mathcal{O}}(\chi) - (\Delta_{\mathcal{O}} + n - \Delta_\phi) F_n^{\mathcal{O}}(\chi). \qquad (2.9)$$

Therefore, the $a^{\text{th}}$ derivative $\partial_\chi^a F_n^{\mathcal{O}}(\chi)$ can be written as

$$\partial_\chi^a F_n^{\mathcal{O}}(\chi) = \sum_m D_{nm}^{(a)}(\Delta_{\mathcal{O}}, \chi) F_m^{\mathcal{O}}(\chi), \qquad (2.10)$$

where $D^{(a)}(\Delta_{\mathcal{O}}, \chi)$ is a banded upper triangular matrix with bandwidth $a$ whose entries depend on $\chi$ and $\Delta_{\mathcal{O}}$.

Thus, the expression obtained by acting with $\partial_{\chi_1}^a \partial_{\chi_2}^b \partial_{\chi_3}^c$ on eq. (2.8) takes the form

$$\sum_{\mathcal{O}} \sum_{m,n=0}^{\infty} \left[ F_m^{\mathcal{O}}(\chi_1) M_{mn}^{(a,b,c)}(\Delta_{\mathcal{O}}, \chi_i) F_n^{\mathcal{O}}(\chi_3) - F_m^{\mathcal{O}}(\chi_2) \tilde{M}_{mn}^{(a,b,c)}(\Delta_{\mathcal{O}}, \chi_i) F_n^{\mathcal{O}}(\chi_4) \right] = 0, \quad (2.11)$$

where $M_{mn}^{(a,b,c)}(\Delta_{\mathcal{O}}, \chi_i)$ and $\tilde{M}_{mn}^{(a,b,c)}(\Delta_{\mathcal{O}}, \chi_i)$ are banded matrices whose bandwidth is bounded above by $\Lambda := a + b + c$ and whose entries depend on $\Delta_{\mathcal{O}}$ and all three cross-ratios.

Finally, we would like to enforce $\chi_1 = \chi_2 = \chi_3 = \chi_4 =: \chi$, which implies $\chi = \frac{1}{3}$ or $\chi = 1$. The constraint $0 < \chi_i < 1$ leads us to exclude $\chi = 1$. The preferred kinematics at which we evaluate the derivative functionals is therefore $\chi_i = \frac{1}{3}$. Evaluating eq. (2.11) at $\chi_i = \frac{1}{3}$ yields the sum rule

$$\sum_{\mathcal{O}} \sum_{m,n=0}^{\infty} F_m^{\mathcal{O}}(\tfrac{1}{3}) M_{mn}^{(a,b,c)}(\Delta_{\mathcal{O}}) F_n^{\mathcal{O}}(\tfrac{1}{3}) = 0 \qquad (2.12)$$

with some banded matrix $M_{mn}^{(a,b,c)}(\Delta_{\mathcal{O}}) = M_{mn}^{(a,b,c)}(\Delta_{\mathcal{O}}, \frac{1}{3}, \frac{1}{3}, \frac{1}{3}) - \tilde{M}_{mn}^{(a,b,c)}(\Delta_{\mathcal{O}}, \frac{1}{3}, \frac{1}{3}, \frac{1}{3})$ of bandwidth $\Lambda \leq a + b + c$. It is straight forward to write down analytic expressions for the matrices $M_{mn}^{(a,b,c)}(\Delta_{\mathcal{O}})$. They can be found in the `Generate_Functionals.nb` file (see Appendix A.1) provided with the arXiv submission of this paper.

## 2.3 The bootstrap

It is straight forward to formulate an SDPs dual to the sum rules (2.12): For each $\Lambda \in \mathbb{Z}_{>0}$ let $\mathcal{F}_\Lambda$ be a basis of $\text{span}(\{M^{(a,b,c)}| \text{ with } a + b + c \leq \Lambda\})$. A spectrum $\mathcal{S}$ of operators

occurring in the triple OPE $\phi \times \phi \times \phi$ is inconsistent with the crossing equation (2.8) if there are coefficients $c_\alpha$ such that

$$\sum_{\alpha \in \mathcal{F}_\Lambda} c_\alpha \alpha(\Delta_\mathcal{O}) \succeq 0 \text{ for all } \mathcal{O} \in \mathcal{S} \quad \text{and} \quad \sum_{\alpha \in \mathcal{F}_\Lambda} c_\alpha \alpha(\Delta_\mathcal{O}) \succ 0 \text{ for at least one } \mathcal{O} \in \mathcal{S}. \quad (2.13)$$

Using essentially standard reasoning of the conformal bootstrap, conditions on the coefficients $c_\alpha$ that lead to bounds on scaling dimensions in $\phi \times \phi \times \phi$ and bounds on the four-point functions $F_n^\mathcal{O}$ were formulated in Reference [26].

The key challenge is then to efficiently search for $c_\alpha$ that have the desired properties. The problem to determine these coefficients is an SDP with a one-parameter family of infinite dimensional semidefinite constraints. To solve it numerically, one has to reduce this problem to an SDP with finitely many finite dimensional semidefinite constraints.

This has been done in Reference [26] by applying the following four steps.

1. Truncate the infinite-dimensional matrices in $\mathcal{F}_\Lambda$ to finite $N \times N$ matrices.

2. Truncate the continuum of semidefinite constraints labelled by $\Delta_\mathcal{O}$ to a finite grid.

3. Solve the truncated SDP for the coefficients $c_\alpha$.

4. If the $c_\alpha$ obtained this way satisfy the full untruncated constraints, stop.
   Else, increase $N$ and consider a denser grid of $\Delta_\mathcal{O}$.

For a successful implementation of this scheme in Reference [26], the banded sparsity structure of the derivative functionals reviewed in Section 2.2 was essential in two ways.

1. For the convergence of the truncation scheme it was crucial that positivity of the functionals at large $\Delta_\mathcal{O}$ and far down the diagonal was imposed by additional semidefinite constraints that supplement the truncated SDP. These asymptotic considerations heavily relied on the banded structure of the functionals.

2. The method applied to determine if a given functional satisfies the full untruncated infinite dimensional constraints consisted of manually constructing sparse matrix decompositions, which reduced the problem to numerically checking positive semidefiniteness for a two-parameter family of finite dimensional matrices.

While the truncation scheme of [26] did successfully produce bounds, the computational effort that was necessary to do so does not compare well with the amount of information contained in them. The truncations, the checks performed after obtaining a candidate functional and the potential necessity to iterate the procedure several times with increasing $N$ and finer grids of scaling dimensions, comes with an undesirably large computational cost. Even more severely, as explained in [26], the need to consider large $N$ of order $10^2$ to $10^3$ makes it necessary to use SDPA instead of SDPB. As already mentioned in the introduction, this is highly problematic since no arbitrary precision parallelisable version of SDPA is available, which implies that bootstrap computations based on SDPA cannot be scaled up to efficiently benefit from the resources provided by large computer clusters.

The procedure that is described in the next section solves these problems by making extensive use of sparse matrix decompositions similar to those that were applied manually in the "checking" step of the truncation scheme of [26]. This simultaneously eliminates the need for any truncations and reduces the six-point bootstrap to SDPs that are very similar to those encountered in the four-point bootstrap. In particular, by using the new formulation, SDPB's capability to efficiently solve such four-point bootstrap problems can directly be applied to the six-point bootstrap.

## 3    New numerics for the six-point bootstrap

This section presents the main technical result of the paper – a new, sparse SDP based formulation of the numerical six-point bootstrap. To this end, Section 3.1 briefly reviews how an SDP for large banded[4] matrices can be reformulated as a system of coupled SDPs for smaller dense matrices. After the general review, Section 3.2 specialises to the specific SDP arising in the six-point bootstrap. Besides improving efficiency of the bootstrap algorithms, we show that exploiting the banded structure makes it straight forward to use polynomial SDP, thereby overcoming the discretisations necessary in [26].

### 3.1    Banded SDP

This section reviews sparsity exploiting decompositions in SDP with banded matrices. We start by setting up such an SDP and some useful notation. Then, the SDP is reformulated in terms of two coupled SDPs of smaller dense matrices by decomposing each of the constraint matrices into an upper left and a lower right part. After this warm-up, a more general decomposition into multiple coupled SDPs is stated. The section ends with a simple example that illustrates the application of the general result.

**Setup and notation.**    For any objective $c \in \mathbb{R}^N$ and any collection of $N + 1$ symmetric $m \times m$ matrices $\{M^\alpha\}_{\alpha=0}^N$, we can use SDP to solve the following optimisation problem.

$$\text{Maximise } c \cdot y \text{ over } y \in \mathbb{R}^N \text{ such that } M^0 + \sum_{\alpha=1}^N y_\alpha M^\alpha \succeq 0. \tag{3.1}$$

In this section, we are interested in the case where $\{M^\alpha\}_{\alpha=0}^N$ are banded matrices with bandwidth $\Lambda$. That is,

$$|i - j| > \Lambda \Rightarrow M_{ij}^\alpha = 0. \tag{3.2}$$

For a discussion of such a sparse SDP, it is useful to introduce $s \times t$ matrices $E_{s,t}^{m_0}$ with

$$(E_{s,t}^{m_0})_{ij} = \delta_{i-j}^{m_0}, \tag{3.3}$$

i.e. $E_{s,t}^{m_0}$ is a matrix that is 1 on a diagonal specified by $m_0$ and 0 otherwise. Note that

$$(E_{s,t}^{m_0})^T = E_{t,s}^{-m_0}. \tag{3.4}$$

Throughout this section, $\text{Sym}_k$ denotes the set of symmetric $k \times k$ matrices.

---

[4]The general theory of sparse SDP is discussed for example in the excellent review [29].

**Splitting the SDP in two.** The matrices defined in eq. (3.3) can be used to conveniently describe projections of matrices onto submatrices. For instance, the upper left $a \times a$ submatrix of the $m \times m$ matrix $M^\alpha$ can be written as

$$A^\alpha := E^0_{a,m} M^\alpha E^0_{m,a}. \tag{3.5}$$

Moreover, $E^{m_0}_{s,t}$ can also be used to embed small matrices into larger matrices. For instance, $E^0_{m,a} A^\alpha E^0_{a,m}$ is a $m \times m$ matrix that coincides with $M^\alpha$ in the upper left $a \times a$ submatrix, but vanishes otherwise. In particular, if we project onto the lower right $b \times b$ submatrix

$$B^\alpha := E^{b-m}_{b,m}(M^\alpha - E^0_{m,a} A^\alpha E^0_{a,m}) E^{m-b}_{m,b}. \tag{3.6}$$

of $M^\alpha - E^0_{m,a} A^\alpha E^0_{a,m}$ then we can recover $M^\alpha$ as

$$E^0_{m,a} A^\alpha E^0_{a,m} + E^{m-b}_{m,b} B^\alpha E^{b-m}_{b,m} = M^\alpha \tag{3.7}$$

for suitably large $a$ and $b$. More concretely, since $M^\alpha$ has bandwidth $\Lambda$, it is enough for $A$ and $B$ to overlap on a $\Lambda \times \Lambda$ submatrix of $M^\alpha$. Because the parts of $M^\alpha$ captured by $A^\alpha$ and $B^\alpha$ overlap in the lower right and upper left $(a+b-m) \times (a+b-m)$ corners respectively, this amounts to the constraint $a + b \geq m + \Lambda$.

The decomposition is of course not unique. For any $X \in \mathrm{Sym}_{a+b-m}$, we can define

$$A^\alpha_X = A^\alpha + E^{m-b}_{a,a+b-m} X E^{b-m}_{a+b-m,a} \qquad B^\alpha_X = B^\alpha - E^0_{b,a+b-m} X E^0_{a+b-m,b} \tag{3.8}$$

and still have the decomposition property

$$E^0_{m,a} A^\alpha_X E^0_{a,m} + E^{m-b}_{m,b} B^\alpha_X E^{b-m}_{b,m} = M^\alpha. \tag{3.9}$$

With this ambiguity in mind, let us consider an optimisation problem with an extended set of variables $y \in \mathbb{R}^N$ and $X \in \mathrm{Sym}_{a+b-m}$, namely

$$\text{Maximise } c \cdot y \text{ such that } A^0_X + \sum_{\alpha=1}^N y_\alpha A^\alpha \succeq 0 \text{ and } B^0_X + \sum_{\alpha=1}^N y_\alpha B^\alpha \succeq 0. \tag{3.10}$$

Clearly, every $y$ that solves the constraints of this SDP also solves the constraint of our original problem (3.1). A less trivial, but very useful fact is that the converse holds too [29]. That is, for every positive semidefinite banded matrix

$$M = M^0 + \sum_{\alpha=1}^N y_\alpha M^\alpha, \tag{3.11}$$

there exists a symmetric matrix $X$ such that

$$A^0 + \sum_{\alpha=1}^N y_\alpha A^\alpha \succeq -E^{m-b}_{a,a+b-m} X E^{b-m}_{a+b-m,a} \tag{3.12}$$

and

$$B^0 + \sum_{\alpha=1}^{N} y_\alpha B^\alpha \succeq E^0_{b,a+b-m} X E^0_{a+b-m,b}. \tag{3.13}$$

Therefore, *the two SDPs* (3.1) *and* (3.10) *are in fact equivalent.* Applying this equivalence iteratively untill we arrive at $m - \Lambda$ coupled SDPs of $(\Lambda+1) \times (\Lambda+1)$ matrices allows us to deduce the following key result.

---

**Dense decomposition of banded SDPs.** *Let $\{M^\alpha\}_{\alpha=0}^N$ be symmetric banded $m \times m$ matrices with bandwidth $\Lambda$ and $c \in \mathbb{R}^N$. Then, the task of maximising $c \cdot y$ over $y \in \mathbb{R}^N$ with the positive semidefinite constraint*

$$M^0 + \sum_{\alpha=1}^{N} y_\alpha M^\alpha \succeq 0 \tag{3.14}$$

*is equivalent to the task of maximising $c \cdot y$ over $(y, X_0, \ldots, X_{m-\Lambda-1}) \in \mathbb{R}^N \times (\mathrm{Sym}_{\Lambda+1})^{m-\Lambda}$ with the constraint*

$$X_n + M_n^0 + \sum_{\alpha=1}^{N} y_\alpha M_n^\alpha \succeq 0 \qquad \sum_{n=0}^{m-\Lambda-1} E^n_{m,\Lambda+1} X_n E^{-n}_{\Lambda+1,m} = 0, \tag{3.15}$$

*where $M_n^\alpha \in Sym_{\Lambda+1}$ are some arbitrary fixed matrices with the property*

$$\sum_{n=0}^{m-\Lambda-1} E^n_{m,\Lambda+1} M_n^\alpha E^{-n}_{\Lambda+1,m} = M^\alpha. \tag{3.16}$$

---

**Example.** As a simple example to illustrate the decomposition discussed in the previous section, consider the problem of determining the maximal $y$ such that

$$M^0 + yM^1 := \begin{pmatrix} 1 & 1 & 0 \\ 1 & 1 & 0 \\ 0 & 0 & 1 \end{pmatrix} + y \begin{pmatrix} 1 & -1 & 0 \\ -1 & 1 & -1 \\ 0 & -1 & 1 \end{pmatrix} \succeq 0. \tag{3.17}$$

This SDP is simple enough to just write down the spectrum of $M^0 + yM^1$, namely

$$\mathrm{spec}(M^0 + yM^1) = \left\{ y + 1, -\sqrt{2y^2 - 2y + 1} + y + 1, \sqrt{2y^2 - 2y + 1} + y + 1 \right\}, \tag{3.18}$$

and simplify $\mathrm{spec}(M^0 + yM^1) \subseteq \mathbb{R}_{\geq 0}$ to conclude that the feasible range of $y$ is

$$0 \leq y \leq 4. \tag{3.19}$$

Alternatively, we can use the decomposition approach discussed in the previous section and conclude that the SDP formulated in eq. (3.17) is equivalent to the problem of finding the maximal $y$ such that there is a pair of symmetric $2 \times 2$ matrices $X_0, X_1$ with

$$X_0 + M_0^0 + yM_0^1 \succeq 0 \qquad X_1 + M_1^0 + yM_1^1 \succeq 0 \qquad E^0_{3,2} X_0 E^0_{2,3} + E^1_{3,2} X_1 E^{-1}_{2,3} = 0, \tag{3.20}$$

where

$$M_0^0 = \begin{pmatrix} 1 & 1 \\ 1 & 1 \end{pmatrix}, \quad M_1^0 = \begin{pmatrix} 0 & 0 \\ 0 & 1 \end{pmatrix}, \quad M_0^1 = \begin{pmatrix} 1 & -1 \\ -1 & 1 \end{pmatrix} \quad \text{and} \quad M_1^1 = \begin{pmatrix} 0 & -1 \\ -1 & 1 \end{pmatrix}. \quad (3.21)$$

The equality constraint is easily solved. Indeed,

$$0 = E_{3,2}^0 X_0 E_{2,3}^0 + E_{3,2}^1 X_1 E_{2,3}^{-1} = \begin{pmatrix} (X_0)_{11} & (X_0)_{12} & 0 \\ (X_0)_{21} & (X_0)_{22} + (X_1)_{11} & (X_1)_{12} \\ 0 & (X_1)_{21} & (X_1)_{22} \end{pmatrix} \quad (3.22)$$

implies

$$X_0 = \begin{pmatrix} 0 & 0 \\ 0 & -x \end{pmatrix} \qquad \text{and} \qquad X_1 = \begin{pmatrix} x & 0 \\ 0 & 0 \end{pmatrix}. \quad (3.23)$$

Thus, our original single variable, $3 \times 3$ matrix SDP has reduced to the system

$$\begin{pmatrix} 1+y & 1-y \\ 1-y & 1+y-x \end{pmatrix} \succeq 0 \qquad \text{and} \qquad \begin{pmatrix} x & -y \\ -y & y+1 \end{pmatrix} \succeq 0 \quad (3.24)$$

of two coupled two variable $2 \times 2$ matrix SDPs. Demanding a non negative trace and determinant for the matrices in eq. (3.24) leads to quadratic inequalities which simplify to

$$0 \le y \le 4 \qquad \text{and} \qquad \frac{y^2}{y+1} \le x \le \frac{4y}{y+1}. \quad (3.25)$$

Projecting to $y$, we recover the feasible interval $0 \le y \le 4$ given in eq. (3.19).

## 3.2 Improved six-point bootstrap

We now apply the sparse SDP techniques reviewed in Section 3.1 to the numerical six-point bootstrap. For this, we first lay out what choices have to be made in the course of the implementation. Then, we detail the specific choices that have been made for this work.

**Overview.** As reviewed in Secs. 2.2 and 2.3, the six-point bootstrap involves constructing linear combinations of infinite dimensional symmetric matrices with bandwidth $\Lambda$, whose entries are rational functions of a conformal weight $\Delta$. Positive semidefiniteness is imposed on the matrices for certain ranges of $\Delta$, e.g. $\Delta > \Delta_*$, that is for all conformal weights above a gap $\Delta_*$. Concretely, the matrices $M^{(a,b,c)}(\Delta)$ defining the sum rules (2.12) take the form

$$M^\alpha(\Delta) = \sum_{n=0}^\infty E_{\infty,\Lambda+1}^n \tilde{M}^\alpha(\Delta, n) E_{\Lambda+1,\infty}^{-n} \qquad \text{with} \qquad \tilde{M}^\alpha(\Delta, n) = \frac{M^\alpha(\Delta, n)}{3^n (2\Delta)_n n!}, \quad (3.26)$$

where the matrices $E$ are defined as in eq. (3.3) and the entries of $M(\Delta, n) \in \mathrm{Sym}_{\Lambda+1}$ are polynomials in $\Delta$ and $n$ of degree

$$\deg_\Delta M_{jk}(\Delta) \le \Lambda + (2 - j - k) \qquad \deg_n M_{jk}(\Delta, n) \le \Lambda + (2 - j - k). \quad (3.27)$$

In particular,

$$j + k > \Lambda + 2 \Rightarrow M_{jk}(\Delta, n) = 0. \tag{3.28}$$

In the spirit of the approach to sparse SDPs laid out in Section 3.1, we reformulate the infinite dimensional semidefinite constraints involving sparse matrices of the type (3.26) as infinite systems of semidefinite constraints for dense $(\Lambda + 1) \times (\Lambda + 1)$ matrices. Eq. (3.26) is already a decomposition of the form (3.16). Let us directly work with this particular choice of breaking the infinite banded matrices up into families of $\text{Sym}_{\Lambda+1}$ matrices. There are two important choices that we need to additionally make. These are

1. *Choice of X matrices.* As in the example (Sec. 3.1) we would like to choose a convenient parametrisation of the constraints on the $X$ matrices in eq. (3.15).

2. *Choice of C matrices.* After we have decomposed the sparse SDP into a coupled family of SDPs for smaller matrices, we can choose a different matrix to conjugate each of the coupled SDPs.

The remainder of this section gives a detailed description of these two choices.

$X$ **matrices.** To describe the construction of $X$ matrices, the following four paragraphs approach the SDPs relevant for numerical six-point bootstrap with four steps of increasing complexity. Concretely, we consider

1. Banded SDPs with matrices whose entries are independent real numbers.

2. Banded SDPs with matrices whose entries are independent polynomials in $\Delta$.

3. Banded SDPs with matrices whose entries are not independent of each other but instead polynomials in two variables, namely $\Delta$ and the position $n$ along the diagonal.

4. The specific SDPs relevant for six-point bootstrap, where matrix elements are, up to an exponential prefactor, rational functions of $\Delta$ and $n$ as indicated in eq. (3.26).

$X$ **matrices, step 1: Constant matrices.** As in the example given at the end of Section 3.1, we would like to find a set of sequences of matrices $\{(X_n^\beta)_{n=0}^\infty\}$ that is a basis of solutions to the constraint

$$\sum_{n=0}^\infty E_{m,\Lambda+1}^n X_n E_{\Lambda+1,m}^{-n} = 0 \tag{3.29}$$

arising in the decomposition of banded SDPs i.e. in eq. (3.15).

One choice of such a basis is given by the sequences $X^{(n_0,I,J)}$ labelled by natural numbers $n_0, I, J$ with $2 \leq I \leq J \leq \Lambda + 1$ and consisting of matrices $X_n^{(n_0,I,J)}$ with entries

$$(X_n^{(n_0,I,J)})_{ij} = \delta_{n_0}^n \delta_I^{(i} \delta_J^{j)} - \delta_{n_0+I-1}^n \delta_1^{(i} \delta_{1+J-I}^{j)} \tag{3.30}$$

It is simple to veryfiy that $X^{(n_0,I,J)}$ indeed satisfies the constraints:

$$\sum_{n=0}^{\infty}(E_{m,\Lambda+1}^n X_n^{(n_0,I,J)} E_{\Lambda+1,m}^{-n})_{ij} = \sum_{n=0}^{\infty}\sum_{k,l=1}^{\Lambda+1}\delta_{i-k}^n (X_n^{(n_0,I,J)})_{kl}\delta_{l-j}^{-n} \tag{3.31}$$

$$= \sum_{n=0}^{\infty}(X_n^{(n_0,I,J)})_{i-n,j-n} = \delta_I^{(i-n_0}\delta_J^{j-n_0)} - \delta_1^{(i-n_0-I+1}\delta_{1+J-I}^{j-n_0-I+1)} = 0. \tag{3.32}$$

Using this basis, we can rephrase the maximisation of $b \cdot y$ over $y$ constrained by

$$M^0 + \sum_{\alpha=1}^{N} y_\alpha M^\alpha \succeq 0 \tag{3.33}$$

with infinite dimensional banded matrices $M^\alpha$ as a maximisation of $b \cdot y$ over two variables $x$ and $y$ constrained by the system

$$M_n^0 + \sum_{p=0}^{\min(\Lambda,n)}\sum_{J=2}^{\Lambda+1}\sum_{I=2}^{J} x_{(n-p,I,J)}X_n^{(n-p,I,J)} + \sum_{\alpha=1}^{N} y_\alpha M_n^\alpha \succeq 0 \qquad \forall n \in \mathbb{N}_0 \tag{3.34}$$

of coupled semidefinite constraints involving only $(\Lambda+1)\times(\Lambda+1)$ matrices.

**X matrices, step 2: Polynomial in $\Delta$ matrices.** It is straight forward to generalise eq. (3.30) to deal with polynomial SDPs of the form

$$\text{maximise } b \cdot y \text{ over } y \text{ such that } M^0(\Delta) + \sum_{\alpha=1}^{N} y_\alpha M^\alpha(\Delta) \succeq 0 \ \forall \Delta \in \mathcal{S}, \tag{3.35}$$

where $\mathcal{S}$ is some subset of $\mathbb{R}$. We simply need to multiply the $X^{(n_0,I,J)}$ with some suitable set of polynomials. Concretely, if we define

$$X^{(n_0,I,J,d_\Delta)} := X^{(n_0,I,J)}\Delta^{d_\Delta}, \tag{3.36}$$

then the SDP in eq. (3.35) is equivalent to

maximise $b \cdot y$ over $(x,y)$ such that

$$M_n^0(\Delta) + \sum_{p=0}^{\min(\Lambda,n)}\sum_{J=2}^{\Lambda+1}\sum_{I=2}^{J}\sum_{d_\Delta=0}^{\deg_\Delta(n-p,I,J)} x_{(n-p,I,J,d_\Delta)}X_n^{(n-p,I,J,d_\Delta)}(\Delta)$$

$$+ \sum_{\alpha=1}^{N} y_\alpha M_n^\alpha(\Delta) \succeq 0 \ \forall n \in \mathbb{N}_0, \Delta \in \mathcal{S}, \tag{3.37}$$

where

$$\deg_\Delta(n_0,I,J) := \max_{0\leq\alpha\leq N}(\max[\deg_\Delta(M_{n_0+I,n_0+J}^\alpha(\Delta)), \deg_\Delta(M_{n_0+1,n_0+1+J-I}^\alpha(\Delta))]). \tag{3.38}$$

**X matrices, step 3: Matrix entries polynomial in $\Delta$ and $n$.** With an infinite number of undetermined coefficients $x_{n,I,J,d_\Delta}$, the SDP formulated in eq. (3.37) seems to be somewhat abstract and not particularly useful for concrete numerical computations. If however, the matrices $M_n^\alpha(\Delta)$ with different $n$ are not independent of each other, but instead have entries that are for instance polynomials $M_{ij}^\alpha(\Delta, n)$ in $n$, we can use eq. (3.37) to formulate a finite dimensional version of the a priori infinite dimensional SDP (3.35).

To do so, we define

$$X_{ij}^{(I,J,d_\Delta,d_n)}(\Delta, n) := \Delta^{d_\Delta}((n + I - 1)^{d_n}\delta_I^{(i}\delta_J^{j)} - n^{d_n}\delta_1^{(i}\delta_{1+J-I}^{j)}H(n - I + 1)), \qquad (3.39)$$

where $H$ is the Heaviside step function

$$H(x) := \begin{cases} 1, & x \geq 0 \\ 0, & x < 0 \end{cases}. \qquad (3.40)$$

Again, it is easy to check that the constraints are satisfied. Indeed,

$$\sum_{n=0}^{\infty}(E_{m,\Lambda+1}^n X^{(I,J,d_\Delta,d_n)}(\Delta, n)E_{\Lambda+1,m}^{-n})_{ij} = \sum_{n=0}^{\infty} X_{i-n,j-n}^{(I,J,d_\Delta,d_n)}(\Delta, n) \qquad (3.41)$$

$$=\Delta^{d_\Delta}\left(\sum_{n=0}^{\infty}(n + I - 1)^{d_n}\delta_I^{(i-n}\delta_J^{j-n)} - \sum_{n=0}^{\infty}n^{d_n}\delta_1^{(i-n}\delta_{1+J-I}^{j-n)}H(n - I + 1)\right) \qquad (3.42)$$

$$=\Delta^{d_\Delta}\left(\sum_{n=0}^{\infty}(n + I - 1)^{d_n}\delta_I^{(i-n}\delta_J^{j-n)} - \sum_{n=0}^{\infty}(n + I - 1)^{d_n}\delta_1^{(i-n-I+1}\delta_{1+J-I}^{j-n-I+1)}\right) = 0. \qquad (3.43)$$

Using the matrices $X^{(I,J,d_\Delta,d_n)}(\Delta, n)$, we can formulate the SDP

maximise $b \cdot y$ over $(x, y)$ such that

$$M^0(\Delta, n) + \sum_{J=2}^{\Lambda+1}\sum_{I=2}^{J}\sum_{d_\Delta=0}^{\deg_\Delta(I,J)}\sum_{d_n=0}^{\deg_n(I,J)} x_{(I,J,d_\Delta,d_n)}X^{(I,J,d_\Delta,d_n)}(\Delta, n)$$

$$+ \sum_{\alpha=1}^{N} y_\alpha M^\alpha(\Delta, n) \succeq 0 \ \forall n \in \mathbb{N}_0, \Delta \in \mathcal{S}, \qquad (3.44)$$

where for both $x = n$ and $x = \Delta$, we define

$$\deg_x(I, J) := \max_{0 \leq \alpha \leq N}(\max[\deg_x(M_{I,J}^\alpha(\Delta, n)), \deg_x(M_{1,1+J-I}^\alpha(\Delta, n))]). \qquad (3.45)$$

In general, this finite SDP has stronger constraints than the original SDP (3.35). Thus, the optimal solution to (3.44) only gives a lower bound for the maximal value of $b \cdot y$ in (3.35). For bootstrap applications this means that, using (3.44), we will still find rigorous bounds on CFT data – however possibly not the optimal bounds. In practice this is not particularly concerning. After all, restricting to a finite number of derivative functionals, we cannot expect optimal bounds in any case.

It is however important to note that we can combine the $X^{(I,J,d_\Delta,d_n)}$ matrices with any choice of finitely many $X^{(n_0,I,J,d_\Delta)}$. The SDPs constructed in that way in principle can be made as close to being equivalent to eq. (3.35) as we like.

**X-matrices step 4: SDPs of the six-point bootstrap.** As already stated earlier in this section, the matrices relevant for six-point bootstrap do not have entries that are polynomial in $\Delta$ and $n$ but rather take the form indicated in equation (3.26). Thus, we cannot directly work with the matrices $X^{(I,J,d_\Delta,d_n)}$ and $X^{(n_0,I,J,d_\Delta)}$. The modified versions that we use instead for numerical bootstrap are

$$\tilde{X}_{ij}^{(I,J,d_\Delta,d_n)}(\Delta,n) = \frac{\Delta^{d_\Delta}}{3^n}\left[\frac{(n+I-1)^{d_n}\delta_I^{(i}\delta_J^{j)}}{3^{I-1}(2\Delta)_{n+I-1}(n+I-1)!} - \frac{n^{d_n}\delta_1^{(i}\delta_{1+J-I}^{j)}H(n-I+1)}{(2\Delta)_n n!}\right] \tag{3.46}$$

$$\text{and} \quad (\tilde{X}_n^{(n_0,I,J,d_\Delta)}(\Delta))_{ij} = \frac{\Delta^{d_\Delta}}{3^n(2\Delta)_n n!}\left(\delta_{n_0}^n\delta_I^{(i}\delta_J^{j)} - \delta_{n_0+I-1}^n\delta_1^{(i}\delta_{1+J-I}^{j)}\right). \tag{3.47}$$

With these, we map the SDP (3.35) with matrices $M^\alpha(\Delta)$ as in (3.26) into the SDP[5]

maximise $b \cdot y$ over $(x,y)$ such that

$$\tilde{M}^0(\Delta,n) + \sum_{\alpha=1}^N y_\alpha \tilde{M}^\alpha(\Delta,n) + \sum_{J=2}^{\Lambda+1}\sum_{I=2}^{J}\sum_{d_n=0}^{\Lambda+I-J}\sum_{d_\Delta=0}^{\Lambda+I-J-d_n} x_{(I,J,d_\Delta,d_n)}\tilde{X}^{(I,J,d_\Delta,d_n)}(\Delta,n)$$

$$+ \sum_{n_0=0}^{\Lambda-1}\sum_{J=2}^{\Lambda+1}\sum_{I=2}^{\min(J,\Lambda-n_0)}\sum_{d_\Delta=0}^{\Lambda+I-J} x_{(n_0,I,J,d_\Delta)}\tilde{X}_n^{(n_0,I,J,d_\Delta)}(\Delta) \succeq 0 \; \forall n \in \mathbb{N}_0, \Delta \in \mathcal{S}. \tag{3.48}$$

As a final comment on the decomposition, let us state the empirical observation that not all of the $X$ matrices have an equal impact on the bootstrap bounds. In particular, some of them do not contribute to the optimal functionals at all. Therefore, we usually only work with a subset of the general matrices described here.

$C$ **matrices.** In order to efficiently solve the SDP formulated in eq. (3.48), one has to use the freedom to conjugate the semidefinite constraints by some sequences of invertible matrices $C_n(\Delta)$. In doing so there are two goals

1. One would like to make sure that after conjugation all semidefinite constraints only involve matrices that depend polynomially on $\Delta$, preferably with a low degree. This allows us to treat the continuous parameter $\Delta$ of the constraints in eq.(3.48) with standard polynomial SDP techniques without truncation.

2. Additionally, one would like the matrix entries of $C_n(\Delta)$ to depend on $n$ in precisely such a way that, after conjugation, the diagonal entries of the constraint matrices are constant to leading order in $n \gg 1$. This allows us to deal with the infinite range of

---

[5]The choice to only include $\tilde{X}_n^{(n_0,I,J,d_\Delta)}(\Delta)$ matrices with $n_0$ up to $\Lambda - 1$ in eq. (3.48) is somewhat arbitrary. As noted towards the end of the description of step 3, one could in principle include a larger number of these matrices. However, the bounds presented in this paper were empirically found not to be sensitive to this choice i.e. including more $\tilde{X}_n^{(n_0,I,J,d_\Delta)}(\Delta)$ matrices than those with $n_0$ up to $\Lambda - 1$ has an effect that is orders of magnitude smaller than increasing $\Lambda$. This aligns well with the observation that in the cases where Reference [26] provided optimal fixed $\Lambda$ bounds to compare with, they agreed with the ones computed with the choice in eq. (3.48), implying that there is no room for improvement.

the $n$ label in eq. (3.48) by imposing positivity in the $n \to \infty$ limit and a finite range of $n$ up to some truncation value[6].

To achieve the first goal, conjugate by the diagonal matrices $C_n^\Delta$ with

$$
(C_n)_{ii} = \begin{cases} \sqrt{3^n (2\Delta)_{n+\lfloor \frac{\Lambda+1}{2} \rfloor} \left(n + \lfloor \frac{\Lambda+1}{2} \rfloor\right)!}, & 0 \le i - 1 \le \lfloor \frac{\Lambda+1}{2} \rfloor \\ \sqrt{\frac{3^n \left(n + \lfloor \frac{\Lambda+1}{2} \rfloor\right)!}{(2\Delta)_{n+\lfloor \frac{\Lambda+1}{2} \rfloor}}} (2\Delta)_{n+i-1}, & \lfloor \frac{\Lambda+1}{2} \rfloor < i - 1 \le \Lambda \end{cases}. \tag{3.49}
$$

To achieve the second goal, additionally conjugate matrices with $n > \Lambda$ by the diagonal matrices $\hat{C}_n$ defined as

$$
(\hat{C}_n)_{ii} = n^{-\deg_n(i)} \qquad \text{where} \qquad \deg_n(i) := \max_\alpha \deg_n(C_n M^\alpha(n\Delta, n)C_n)_{ii}. \tag{3.50}
$$

## 4    Extremal flow from GFF to GFB

This section uses the improved numerical six-point bootstrap to study extremal correlators that interpolate between the Generalised Free Fermion (GFF) and the Generalised Free Boson (GFB).

A similar four-point bootstrap analysis was performed in Reference [28]. There, a one-parameter family of OPE maximising solutions to four-point crossing that interpolates between GFB and GFF was studied using the numerical bootstrap. Close to GFB, it was related to the deformation of the massive free boson in $\mathrm{AdS}_2$ by a quartic interaction. Here, this one-parameter family of extremal correlators is extended to a two-parameter family incorporating also the sextic interaction. The logic of this six-point extension follows similar bootstrap problems studied in Reference [26] where the CFT data contained in the four-point function of four identical generalised free bosons or fermions was used as an input to study the constraints that six-point crossing puts on additional CFT data occurring in the respective six-point functions.

As a warm-up, Section 4.1 briefly showcases the consequences of the improvements that were presented in Section 3 by revisiting the "gap maximisation without identity" problem of [26]. Section 4.2 then turns to the extremal four-point correlators interpolating between GFB and GFF. In Section 4.3, a six-point bootstrap problem that produces bounds on the triple twist data along the four-point deformation is introduced and the results of its implementation are discussed. Finally, Section 4.4 relates these results to perturbative computations in $\mathrm{AdS}_2$.

---

[6]The truncation in $n$ is completely analogous to the standard truncation of the spin parameter in higher dimensional four-point bootstrap. In this work, only fully converged bounds for which the truncation parameter was chosen large enough to ensure that the functionals are positive for all $n$ are presented.

## 4.1 Warm-up: Gap maximisation without identity revisited

The first application of numerical six-point bootstrap studied in Reference [26] was "gap maximisation without identity" i.e. the problem of maximising the dimension of the leading operator in the $\phi \times \phi \times \phi$ triple OPE.

> **Gap maximisation without identity.** *For fixed* $\Delta_\phi$, *determine the maximal* $\Delta_*$ *such that there is a crossing symmetric six-point function* $\langle \phi\phi\phi\phi\phi\phi \rangle$ *consistent with unitarity and the assumption that*
>
> $$\phi \times \phi \times \phi = \phi_* + \ldots, \tag{4.1}$$
>
> *where the operators that have been omitted have a scaling dimension* $\Delta > \Delta_*$.

In Reference [26] an explicit candidate for the extremal solution to gap maximisation without identity was proposed. The proposed correlator has

$$\frac{\Delta_*}{\Delta_\phi} = \frac{9}{5}. \tag{4.2}$$

The qualifier "without identity" is motivated by the fact that the identity operator neither appears in the $\phi \times \phi \times \phi$ nor in the $\phi \times \phi$ OPE of this extremal solution (the former would mean $\Delta_* = 0$, while the latter would imply $\Delta_* = \Delta_\phi$). Some initial numerical data supporting the claim that the correlator associated with eq. (4.2) is the optimal solution to gap maximisation without identity was already presented in Reference [26].

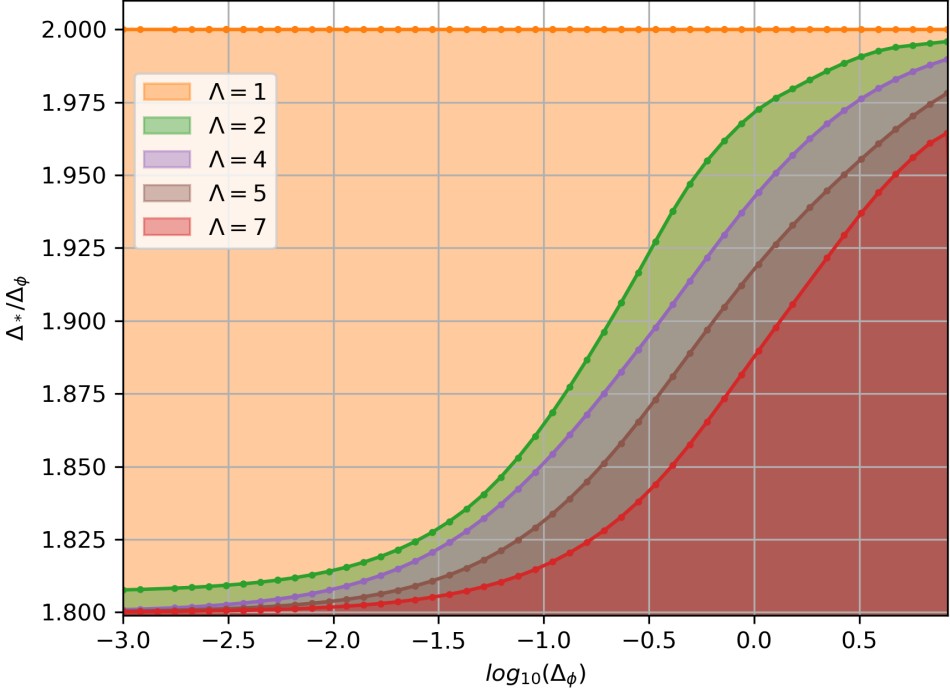

**Figure 2**: Allowed region for gap maximisation without identity.

The methods introduced in Section 3 enable a significant increase of the number of derivative functionals: While the numerical data of the previous publication only contained bounds with up to $\Lambda = 4$ derivatives (i.e. $N_\alpha = 15$ functionals), Figure 2 now shows the bounds for values of $\Lambda$ up to 7 i.e. $N_\alpha = 53$. Apart from increasing $\Lambda$, the new numerical implementation also helped to increase the range of external dimensions $\Delta_\phi$ that could be scanned over. Thanks to these two improvements, a clear picture arises: For small $\Delta_\phi$ the bound rapidly converges to the expected gap of $\frac{9}{5}\Delta_\phi$. For any fixed $\Lambda$, the bound then weakens and monotonically tends to $2\Delta_\phi$ at large $\Delta_\phi$. As $\Lambda$ is increased the region where this transition happens is pushed to larger and larger $\Delta_\phi$.

Appendix A.2 describes some code shared in the anxilliary files of this publication that reproduces the bound presented in Figure 2.

## 4.2 Double twist data from four-point bootstrap

Let us now state the four-point bootstrap problem whose six-point generalisation is the main target of this section and briefly discuss its solution.

> **Three-point maximisation.** For fixed $\Delta_\phi$ and $\Delta_{\phi^2}$, maximise the OPE coefficient $C_{\phi\phi\phi^2}$ such that $\langle\phi\phi\phi\rangle$ is consistent with crossing, unitarity and the assumption that
>
> $$\phi \times \phi = \mathbf{1} + \phi^2 + \dots, \tag{4.3}$$
>
> where the operators that have been omitted have a scaling dimension $\Delta > \Delta_{\phi^2}$.

The $\Delta_{\phi^2} = 2\Delta_\phi$ extremal solution to the three-point maximisation problem is the GFB four-point function. For $\Delta_{\phi^2} = 2\Delta_\phi + 1$, one recovers the GFF [28]. Scaling dimensions and OPE coefficients of the one-parameter family of solutions to crossing interpolating between these two end points are most conveniently inferred by numerically computing solutions to truncated crossing equations that approximate the GFF (or GFB) and applying Newton's method to gradually deform these solutions by varying $\Delta_{\phi^2}$. While it was observed in Reference [28] that this is most elegantly done using analytic functionals, it is sufficient for our purposes to simply compute the truncated spectra for up to $\Lambda_4 = 20$ derivative functionals combined with extrapolation to $\Lambda_4 = \infty$.

Since [28] already gave a detailed description of the spectrum of the extremal correlators, we refrain from further discussing the well-known four-point data and instead directly move on to the new information that can be extracted from the six-point bootstrap. The reader interested in more details on the four-point data is invited to inspect the mathematica notebook `GFF_to_GFB.nb` which we used to generate this data and which is provided in the ancillary files associated to this publication.

## 4.3 Triple twist data from six-point bootstrap

Analogously to the three-point maximisation problem formulated in the previous section, one can study an extended six-point bootstrap problem that maximises four-point functions.

> **Four-point maximisation.** *For fixed $\Delta_\phi$, $\Delta_{\phi^2}$ and $\Delta_{\phi^3}$, maximise the value of the four-point function $F_{\phi^3} = \langle\phi(\infty)\phi(1)\phi^3(1/3)\phi(0)\rangle$ with the constraint that $\langle\phi\phi\phi\phi\phi\rangle$ is consistent with crossing, unitarity and the assumption that*
>
> $$\langle\phi\phi\phi\phi\rangle = F_\phi^{\Delta_{\phi^2}} \qquad \text{and} \qquad \phi\times\phi\times\phi = \phi + \phi^3 + \ldots, \qquad (4.4)$$
>
> *where the operators that have been ommitted have a scaling dimension $\Delta > \Delta_{\phi^3}$.*

Here, $F_\phi^{\Delta_{\phi^2}}$ denotes the solution to three-point maximisation for fixed $\Delta_{\phi^2}$. Upper bounds on the solutions to four-point maximisation can be constructed through the procedure outlined in Section 4.3. of Reference [26]. In fact, that paper already presented numerical data strongly suggesting that for $(\Delta_{\phi^2}, \Delta_{\phi^3}) = (2\Delta_\phi + 1, 3\Delta_\phi + 3)$, where the four-point input $F_\phi^{2\Delta_\phi+1}$ is just the GFF four-point correlator, the four-point maximisation problem is solved by the GFF six-point function. We change the procedure used to obtain this result in two ways here.

1. We use the improved six-point bootstrap described in Section 3.

2. As input data, we use the $\phi$-exchange contribution to the six-point function computed using the extremal solutions to the three-point maximisation problem i.e. if

$$F_\phi^{\Delta_{\phi^2}} = \sum_{\mathcal{O}\in(\phi\times\phi)_{\Delta_{\phi^2}}} C_{\phi\phi\mathcal{O}}^2(\Delta_{\phi^2})G_{\mathcal{O}}, \qquad (4.5)$$

   then the input for the six-point bootstrap is

$$\sum_{\mathcal{O}_1,\mathcal{O}_2\in(\phi\times\phi)_{\Delta_{\phi^2}}} C_{\phi\phi\mathcal{O}_1}^2(\Delta_{\phi^2})C_{\phi\phi\mathcal{O}_2}^2(\Delta_{\phi^2})G_{\mathcal{O}_1\phi\mathcal{O}_2}, \qquad (4.6)$$

   where $G_{\mathcal{O}_1\phi\mathcal{O}_2}$ is the $(\mathcal{O}_1\phi\mathcal{O}_2)$-exchange comb-channel six-point block [30, 31].

For fast, precise evaluation of derivatives comb-channel conformal blocks that enters in eq. (4.6), we use a C programme that together with a python and mathematica wrapper is available in the ancillary files provided with this publication (see also Appendix A.4).

The numerical upper bound for the cases $\Delta_\phi = 0.01$ and $\Delta_\phi = 0.1$ is shown in figures 3 and 4 respectively. For $(\Delta_{\phi^2}, \Delta_{\phi^3}) = (2\Delta_\phi, 3\Delta_\phi)$, the bound is well saturated by the GFB correlator. Likewise, it is saturated by the GFF for $(\Delta_{\phi^2}, \Delta_{\phi^3}) = (2\Delta_\phi + 1, 3\Delta_\phi + 3)$. As a function of $\Delta_{\phi^3}$ at fixed $\Delta_{\phi^2}$, the bound converges rapidly close to the maximal allowed value of $\Delta_{\phi^3}$. As $\Delta_{\phi^3}$ is decreased the bound monotonically increases on the GFB slice $\Delta_{\phi^2} = 2\Delta_\phi$. Closer to the GFF slice $\Delta_{\phi^2} = 2\Delta_\phi + 1$, however, it first drops to a local minimum before finally rising again. This qualitative behaviour does not greatly vary as the value of $\Delta_\phi$ is changed.

## 4.4 Matching with perturbation theory in AdS$_2$

As pointed out in Section 4.3, the GFB/GFF correlators (dual to a massive free boson/fermion in AdS$_2$) saturate the six-point bootstrap bounds and are therefore extremal

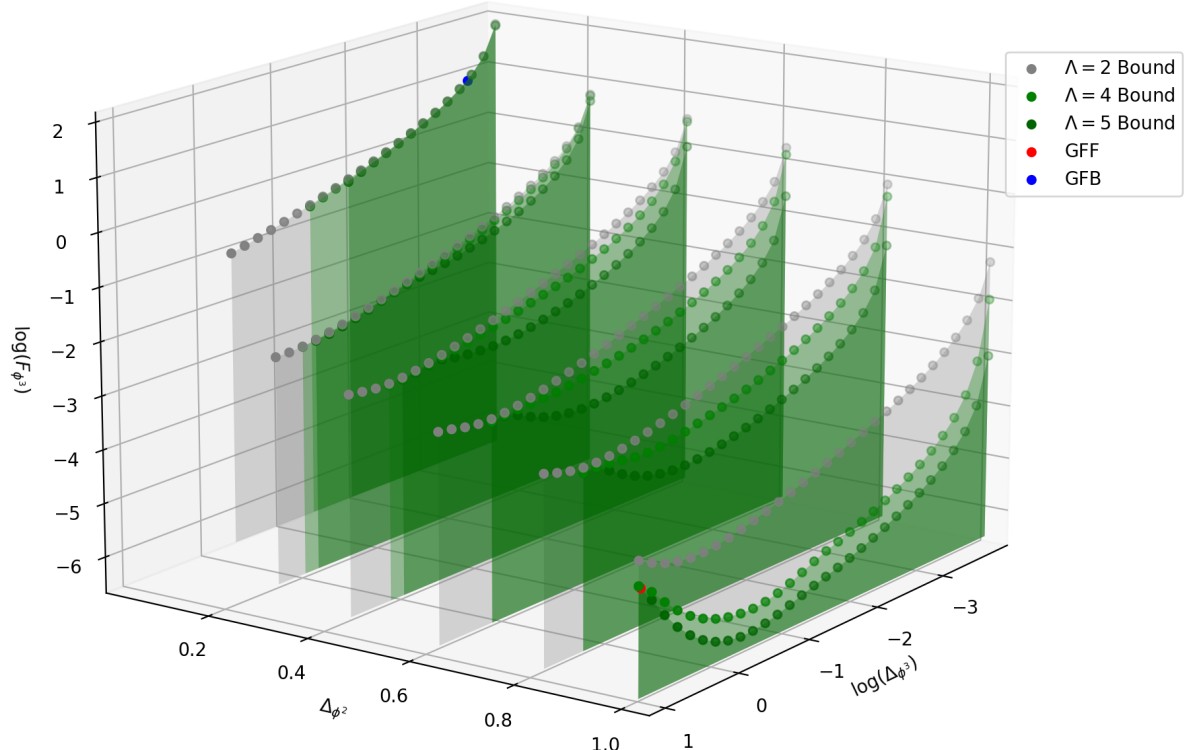

**Figure 3**: Upper bound on solutions of the four-point maximisation problem for $\Delta_\phi = 0.01$.

correlators. A natural expectation is that deformations which preserve extremality, i.e. generate the extremal surface, correspond to interactions that do not introduce new states in the OPE, but instead just modify the dimensions and OPE coefficients of states that are already present. At leading order in a quartic or sextic deformation of the GFB, it is easy to see that no new states are introduced. Therefore, we focus on these examples and relate them to the bounds obtained in Section 4.3.

**$\Phi^6$ interaction.** Let us start from the UV action

$$S = \frac{1}{2}\int d^2 x \sqrt{g}\left(\partial_\mu \Phi \partial^\mu \Phi + m^2 : \Phi^2 : + \frac{\lambda_6}{6!} : \Phi^6 :\right), \tag{4.7}$$

where $\lambda_6 = 0$ describes a free boson $\Phi$ of mass $m^2 = \Delta_\phi(\Delta_\phi - 1)$ dual to a GFB $\phi$ of dimension $\Delta_\phi$. Normal ordering in the sextic interaction by which we deform the boson ensures that no effective quartic interaction is induced at tree level. Therefore, the boundary four-point function is unaltered by the deformation at leading order. That is,

$$\langle \phi(x_1)\phi(x_2)\phi(x_3)\phi(x_4)\rangle = \langle \phi(x_1)\phi(x_2)\phi(x_3)\phi(x_4)\rangle_{\text{GFB}} + O(\lambda_6^2) \tag{4.8}$$

which means that, at this order, the sextic interaction takes us away from the GFB point in Figures 3 and 4 by moving only in the $\Delta_{\phi^3}$ direction while keeping $\Delta_{\phi^2}$ fixed. To make contact with the bounds of Section 4.3, we not only need to compute tree-level corrections to the four-point function $\langle \phi\phi\phi\phi^3\rangle$, but also the anomalous dimension $\gamma_{\phi^3} = \Delta_{\phi^3} - 3\Delta_\phi$

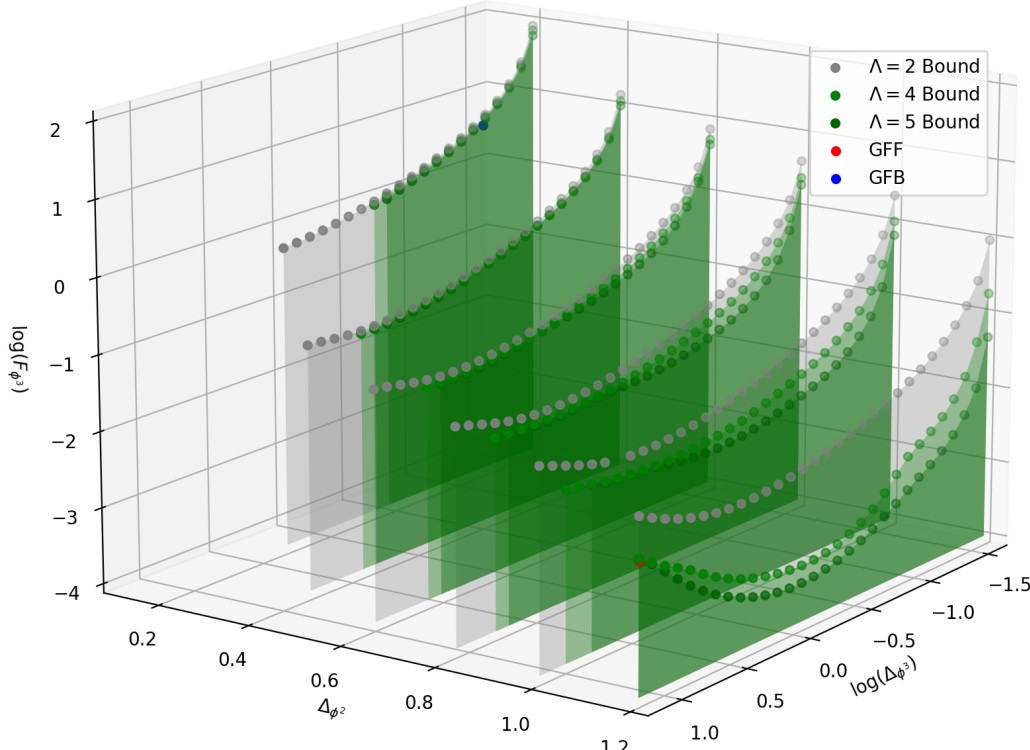

**Figure 4**: Upper bound on solutions of the four-point maximisation problem for $\Delta_\phi = 0.1$.

of $\phi^3$. The later quantifies the rate at which we move away from the GFB point in the $(\Delta_{\phi^2}, \Delta_{\phi^3})$ plane.

To determine $\gamma_{\phi^3}$, consider the correlator

$$\langle\phi(x_1)\phi^2(x_2)\phi(x_3)\phi^2(x_4)\rangle = \text{GFB} - \lambda_6 N_\phi^6 D_{\Delta_\phi, 2\Delta_\phi, \Delta_\phi, 2\Delta_\phi}(x_i) + O(\lambda_6^2) \qquad (4.9)$$

where $D_{\Delta_\phi 2\Delta_\phi \Delta_\phi 2\Delta_\phi}(x_i)$ denotes a D-function, i.e. a quartic contact Witten-diagram and $N_\phi$ is the normalisation of the bulk-to-boundary propagator associated to the diagram. The diagram is drawn on the l.h.s. of Figure 5. Concerning $D$ functions, we use the convention

$$D_{\Delta_1, \Delta_2, \Delta_3, \Delta_4}(x_i) = \int d^2x \sqrt{g} K_{\Delta_1}(x, x_1) K_{\Delta_2}(x, x_2) K_{\Delta_3}(x, x_3) K_{\Delta_4}(x, x_4), \qquad (4.10)$$

where $K_{\Delta_i}(x, x_i)$ is the bulk-to boundary propagator from the point $x$ in the bulk to the point $x_i$ on the boundary. In Poincaré coordinates for AdS$_2$, the bulk-to-boundary propagator reads

$$K_{\Delta_i}(x, x_i) = \left(\frac{y}{y^2 + (x - x_i)^2}\right)^{\Delta_i}. \qquad (4.11)$$

Crucially, eq. (4.11) makes the fact that the product of two bulk-to-boundary propagators of dimension $\Delta_\phi$ is proportional to a single propagator of dimension $2\Delta_\phi$ manifest. This justifies the subscripts of the D-function in eq. (4.9).

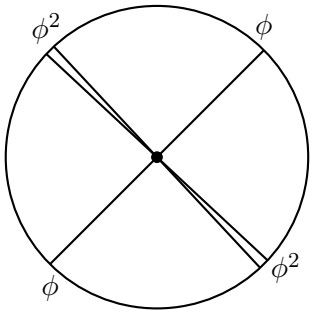
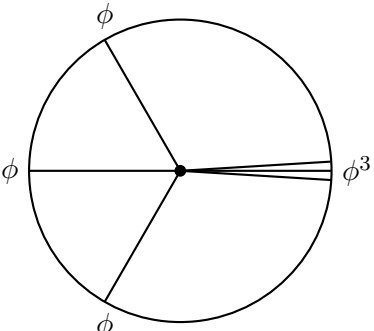

(a) Sextic contact diagram used to determine the anomalous dimension $\gamma_{\phi^3}$ of the triple trace operator $\phi^3$.

(b) Sextic contact diagram leading to a correction to the correlator $\langle\phi\phi\phi\phi^3\rangle$. The diagram is naively divergent.

**Figure 5**: Tree level Witten diagrams occurring in $\lambda_6\Phi^6$ perturbation theory.

Reading off the correction to CFT data from the conformal block expansion of the contact diagram is a standard computation described for example in [32]. For generic external dimensions, the $s$-channel block expansion of the contact diagram contains only double trace towers of the channel pairs, i.e.

$$D_{\Delta_1,\Delta_2,\Delta_3,\Delta_4} = \sum_{n=0}^{\infty} a_n^{12} G_{\Delta_1+\Delta_2+2n}^{12,34} + \sum_{n=0}^{\infty} a_n^{34} G_{\Delta_3+\Delta_4+2n}^{12,34}, \tag{4.12}$$

with the product of OPE coefficients $a_n^{12}$ given by

$$a_n^{12} = \frac{\sqrt{\pi}(-1)^{-n}\Gamma\left(n+\Delta_1\right)\Gamma\left(n+\Delta_2\right)\Gamma\left(n+\Delta_1+\Delta_2-\frac{1}{2}\right)\Gamma\left(\frac{2n+\Delta_1+\Delta_2+\Delta_3-\Delta_4}{2}\right)}{2n!\Gamma\left(\Delta_1\right)\Gamma\left(\Delta_2\right)\Gamma\left(\Delta_3\right)\Gamma\left(\Delta_4\right)}$$
$$\frac{\Gamma\left(\frac{2n+\Delta_1+\Delta_2-\Delta_3+\Delta_4}{2}\right)\Gamma\left(\frac{-2n-\Delta_1-\Delta_2+\Delta_3+\Delta_4}{2}\right)\Gamma\left(\frac{2n+\Delta_1+\Delta_2+\Delta_3+\Delta_4-1}{2}\right)}{\Gamma\left(2n+\Delta_1+\Delta_2-\frac{1}{2}\right)\Gamma\left(2n+\Delta_1+\Delta_2\right)}. \tag{4.13}$$

Considering the first order piece of eq. (4.9) and extracting a dimensionful prefactor

$$\langle\phi(x_1)\phi^2(x_2)\phi(x_3)\phi^2(x_4)\rangle^{(1)} = \frac{1}{x_{12}^{3\Delta_\phi}x_{34}^{3\Delta_\phi}}\left(\frac{x_{13}}{x_{24}}\right)^{\Delta_\phi}f_{\phi\phi^2\phi\phi^2}(z), \tag{4.14}$$

one has the conformal block expansion

$$f_{\phi\phi^2\phi\phi^2}(z) = \sum_{n=0}^{\infty}(C^2_{\phi\phi^2(\phi^3)_n})^{(1)}G_{(\phi^3)_n}^{\phi\phi^2\phi\phi^2}(z) + \sum_{n=0}^{\infty}(C^2_{\phi\phi^2(\phi^3)_n})^{(0)}\gamma_{(\phi^3)_n}^{(1)}\partial_\Delta G_{(\phi^3)_n}^{\phi\phi^2\phi\phi^2}(z), \tag{4.15}$$

which is obtained from the generic expression in the case where $\Delta_1+\Delta_2 = \Delta_3+\Delta_4$. In this case the individual coefficients $a_n^{12}$ and $a_n^{34}$ in eq. (4.12) diverge but the sum of the two towers of double trace operators is finite. Through the computation of this finite limit, derivatives of conformal blocks enter on the r.h.s. of eq. (4.15). The anomalous dimension of $\phi^3$ can be read off as the coefficient which multiplies the derivative of the block corresponding to

the non-degenerate state at $n = 0$. Dividing by the free OPE coefficient $(C^2_{\phi\phi^2(\phi^3)_n})^{(0)} = 3$ and by a factor of 2 ensuring unit normalisation of the $\phi^2$ operators, one finds

$$\gamma_{\phi^3} = -\lambda_6 N_\phi^6 \left( -\frac{\sqrt{\pi}\,\Gamma\left(3\Delta_\phi - \frac{1}{2}\right)}{6\,\Gamma(3\Delta_\phi)} \right). \tag{4.16}$$

We can also read off the OPE coefficient $C^{(1)}_{\phi\phi^2\phi^3}$ by noting that the square OPE coefficient appearing in the correlator is $(C^2_{\phi\phi^2\phi^3})^{(1)} = 2C^{(0)}_{\phi\phi^2\phi^3}C^{(1)}_{\phi\phi^2\phi^3}$. We find

$$C^{(1)}_{\phi\phi^2\phi^3} = \lambda_6 N_\phi^6 \frac{\sqrt{\frac{\pi}{3}}\,\Gamma\left(3\Delta_\phi - \frac{1}{2}\right)\left[H_{\Delta_\phi-1} + H_{2\Delta_\phi-1} + H_{3\Delta_\phi+\frac{3}{2}} - 2H_{6\Delta_\phi-2} + \log(4)\right]}{4\Gamma\left(3\Delta_\phi\right)}, \tag{4.17}$$

where $H_\alpha$ is an analytically continued harmonic number.

Having computed the anomalous dimension, let us turn to the four-point function $f_{\phi\phi\phi\phi^3}(z = 1/3)$, which is the quantity that was bounded in Section 4.3 by the numerical bootstrap. The tree level correction to this correlator is again given by a contact diagram

$$\langle\phi(x_1)\phi(x_2)\phi(x_3)\phi^3(x_4)\rangle^{(1)} = -\lambda_6 N_\phi^6 D_{\Delta_\phi\Delta_\phi\Delta_\phi 3\Delta_\phi}(x_i). \tag{4.18}$$

However this D-function, illustrated on the r.h.s. of Fig. 5, is actually divergent. The divergence is due to the extremality condition $\Delta_\phi + \Delta_\phi + \Delta_\phi = 3\Delta_\phi$, and is a familiar phenomenon in holographic three point-functions. Physically, the extremality condition should be avoided by the fact that the external operator $\phi^3$ acquires an anomalous dimension. Therefore, we regulate the divergence by working in terms of physical CFT data. Let us first remove a kinematic prefactor and define a function $f^{(1)}_{\phi\phi\phi\phi^3}$ of the cross-ratio through

$$\langle\phi(x_1)\phi(x_2)\phi(x_3)\phi^3(x_4)\rangle^{(1)} = \frac{1}{x_{12}^{2\Delta_\phi} x_{34}^{\Delta_\phi+\Delta_{\phi^3}}}\left(\frac{x_{13}}{x_{14}}\right)^{\Delta_{\phi^3}-\Delta_\phi} f^{(1)}_{\phi\phi\phi\phi^3}(z). \tag{4.19}$$

The decomposition of $f^{(1)}_{\phi\phi\phi\phi^3}$ into conformal blocks takes the form

$$f^{(1)}_{\phi\phi\phi\phi^3}(z) = \sum_{n=0}^{\infty} C^{(1)}_{\phi\phi(\phi^4)_n}C^{(0)}_{\phi\phi^3(\phi^4)_n}G^{\phi\phi\phi\phi^3}_{(\phi^4)_n} + \sum_{n=0}^{\infty} C^{(0)}_{\phi\phi(\phi^2)_n}C^{(1)}_{\phi\phi^3(\phi^2)_n}G^{\phi\phi\phi\phi^3}_{(\phi^2)_n}$$
$$+ C^{(0)}_{\phi\phi\phi^2}C^{(0)}_{\phi\phi^3\phi^2}\gamma^{(1)}_{\phi^3}\partial_{\phi^3}G^{\phi\phi\phi\phi^3}_{\phi^2}. \tag{4.20}$$

Clearly, the anomalous dimension of $\phi^3$, which should be unit normalised by diving by $\sqrt{6}$, only affects the exchange of the operator $\phi^2$. This is consistent with the fact that the free piece of the correlator actually contains only a single conformal block

$$f^{(0)}_{\phi\phi\phi\phi^3}(z) = \sqrt{6}\,G^{\phi\phi\phi\phi^3}_{\phi^2}(z). \tag{4.21}$$

Since therefore only the coefficient of the $\phi^2$ block is divergent when using the standard expansion of the contact diagram, we can extract all the coefficients for exchanges of $(\phi^4)_n$ and $(\phi^2)_n$ with $n \geq 1$ from the standard expansion of the D-function. To reconstruct

the $\phi^2$ exchange, we use the expressions (4.16) and (4.17) for $\gamma_{\phi^3}$ and $C^{(1)}_{\phi\phi^2\phi^3}$ that were extracted from the finite $\langle\phi\phi^2\phi\phi^2\rangle$ correlator. Since the OPE converges quickly at $z = 1/3$, it is sufficient to sum only a few blocks to obtain an answer that has converged to several decimal places.

The result can be matched to the derivative of the numerical bound $F_{\phi^3}(\Delta_\phi, \Delta_{\phi^2}, \Delta_{\phi^3})$ w.r.t. $\Delta_{\phi^3}$. As Figure 6 shows, we indeed find

$$\partial_{\Delta_{\phi^3}} F_{\phi^3}|_{GFB} = \frac{\partial_{\lambda_6} F_{\phi^3}}{\partial_{\lambda_6} \Delta_{\phi^3}} = \frac{f^{(1)}_{\phi\phi\phi\phi^3}}{\gamma^{(1)}_{\phi^3}} \tag{4.22}$$

to high precision at small $\Delta_\phi$.

As already pointed out in Section 4.1, the bounds obtained using six-point derivative functionals behave qualitatively similar to those obtained using derivative functionals in the four-point bootstrap, in that the convergence of the bound as a function of the number of functionals significantly slows down as the external dimension $\Delta_\phi$ is increased. To see the effect of this slow down also for the problem studied here, the range of $\Delta_\phi$ shown in Figure 6 also includes a region where the numerical bound is not yet well converged. In that region, the numerical answer and the analytic prediction do of course not agree, it is however reassuring that increasing $\Lambda$ from 4 to 5 reduces the mismatch. This suggests that eventually, the numerical data should match the analytic curve also for large values of $\Delta_\phi$ if a sufficiently high value of $\Lambda$ is chosen.

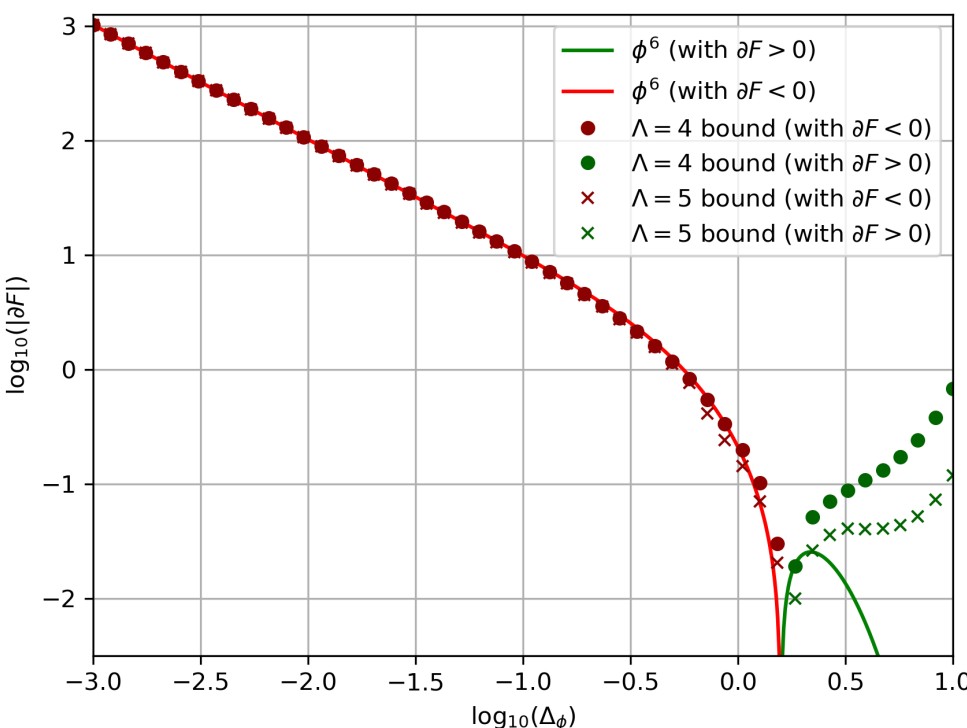

**Figure 6**: Derivative of the four-point function w.r.t. the $\Phi^6$ coupling and numerical derivative of the bound at GFB as a function of $\Delta_\phi$.

$\Phi^4$ **interaction.** The quartic coupling case is similar but has some important differences. We once again normal order the interaction and write the action as

$$S = \frac{1}{2} \int d^2x \sqrt{g} \left( \partial_\mu \Phi \partial^\mu \Phi + m^2 : \Phi^2 : + \frac{\lambda_4}{4!} : \Phi^4 : \right) . \tag{4.23}$$

This of course means that the four-point function of $\phi$ gets modified by a contact diagram

$$\langle \phi(x_1)\phi(x_2)\phi(x_3)\phi(x_4) \rangle = \text{GFF} - \lambda_4 N_\phi^4 D_{\Delta_\phi \Delta_\phi \Delta_\phi \Delta_\phi}(x_i) + O(\lambda_4^2), \tag{4.24}$$

which in particular induces an anomalous dimension for $\phi^2$

$$\gamma_{\phi^2} = -\lambda_4 N_\phi^4 \left( -\frac{\sqrt{\pi}\Gamma\left(2\Delta_\phi - \frac{1}{2}\right)}{2\Gamma\left(2\Delta_\phi\right)} \right), \tag{4.25}$$

and a correction to the OPE coefficient

$$C^{(1)}_{\phi\phi\phi^2} = (-\lambda_4 N_\phi^4) \frac{-\sqrt{\frac{\pi}{2}}\Gamma\left(2\Delta_\phi - \frac{1}{2}\right)\left(2H_{\Delta_\phi - 1} + H_{2\Delta_\phi - \frac{3}{2}} - 2H_{4\Delta_\phi - 2} + \log(4)\right)}{2\Gamma\left(2\Delta_\phi\right)} . \tag{4.26}$$

To obtain the anomalous dimension of $\phi^3$ we could once again study the correlator $\langle \phi \phi^2 \phi \phi^2 \rangle$, but we will instead consider a shortcut. Directly computing the two-point function of $\phi^3$ perturbatively makes it clear that the resulting integral is the same as in the $\phi^2$ case. Counting diagrams leads to a simple combinatorial factor

$$\gamma_{\phi^3} = 3\gamma_{\phi^2} . \tag{4.27}$$

Finally, we need to compute $\langle \phi\phi\phi\phi^3 \rangle$. Since the $\phi^3$ operator is normal ordered, there is no contribution to the correlator that connects all four-points. Instead, the correlator factorises into a product of two- and three-point functions, as illustrated in Figure 7. However these three-point functions are extremal (due to the property $\Delta_\phi + \Delta_\phi = 2\Delta_\phi$) and are therefore divergent. We once again regulate by working in terms of physical data. The divergent three-point integral computes the correction to the three-point function. Therefore, we simply write down the three-point function in terms of OPE coefficients and scaling dimensions and expand them to first order. These are known from the expansion of the finite four-point function $\langle \phi\phi\phi\phi \rangle$ that we wrote above. We then have

$$\langle \phi^2(x_1)\phi(x_2)\phi(x_3) \rangle^{(1)} = \frac{C^1_{\phi\phi\phi^2} - \sqrt{2}\gamma_{\phi^2}(\log(x_{12}x_{13}x_{23}^{-1}))}{x_{12}^{2\Delta_\phi}x_{13}^{2\Delta_\phi}} . \tag{4.28}$$

Adding all the permutations of the above diagram and extracting the prefactor yields a correlator that is easy to write as an explicit function of $z$

$$f^{(1)}_{\phi\phi\phi\phi^3}(z) = -\lambda_4 N_\phi^4 \left( \sqrt{3}z^{2\Delta_\phi}(3C^{(1)}_{\phi\phi\phi^2} + \sqrt{2}\gamma_{\phi^2}\log(z(1-z))) \right), \tag{4.29}$$

which can be straightforwardly evaluated at $z = 1/3$.

Figure 8 visualises numerical data indicating that the $\Phi^4$ deformation is tangential to bound in the same way as the $\Phi^6$ deformation. As opposed to Figure 6, this plot is restricted

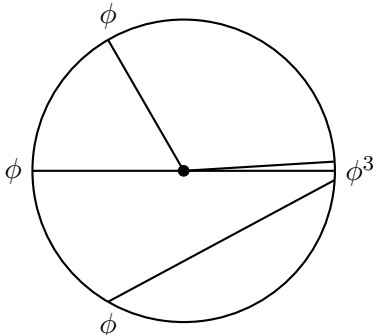

**Figure 7**: Factorised diagram contributing to the $\langle \phi\phi\phi\phi^3 \rangle$ correlator with a quartic vertex. The sub-diagram corresponding to the three-point function is divergent.

to the small $\Delta_\phi$ range where the bound is well converged at $\Lambda = 4$. Similar comments to those made about convergence above Figure 6 also hold here.

Note that, since now both $\Delta_{\phi^2}$ and $\Delta_{\phi^3}$ are changed by the deformation, a combination of the two derivatives has to be taken and we find

$$3\partial_{\Delta_{\phi^3}} F_{\phi^3}|_{GFB} + \partial_{\Delta_{\phi^2}} F_{\phi^3}|_{GFB} = \frac{f^{(1)}_{\phi\phi\phi\phi^3}}{\gamma^{(1)}_{\phi^2}}. \tag{4.30}$$

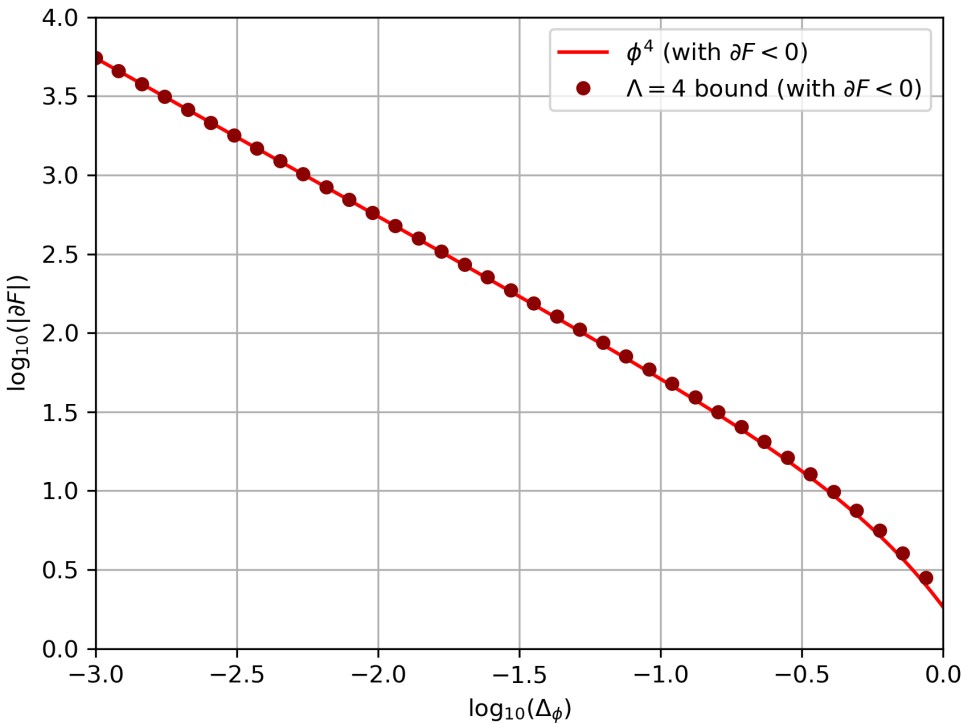

**Figure 8**: Derivative of the four-point function w.r.t. the $\Phi^4$ coupling and numerical derivative of the bound at GFB as a function of $\Delta_\phi$.

## 5 Conclusion

This work introduced sparse SDP techniques to the numerical six-point bootstrap, resulting in significant performance gains and making the many advantages of the conformal bootstrap's standard SDP solver `SDPB` accessible in multi-point applications. While this improvement constitutes a key step in the technical advancement of higher-point numerical bootstrap, there are stil many important challenges and necessary future steps ahead for the development of this programme. Let us end by briefly describing some of these.

**Six-point bootstrap in two dimensions.** The restriction to 1d poses a significant limitation to the applicability of the numerical methods described in this paper. It would therefore be highly desirable to extend the present formalism to higher dimensional CFTs. Formulating a numerical six-point bootstrap in 2d seems to be the most natural realistic first step in this direction. Such a generalisation would increase the set of discrete parameters from the descendant level $n$ encountered in 1d to the two independent levels $n, \bar{n}$ and the primary spin $J$ encountered in 2d. Importantly, taking derivatives w.r.t. the holomorphic and antiholomorphic cross-ratios available in 2d preserves the band structure that was crucial to the success of the 1d six-point bootstrap. It therefore seems plausible that, while numerically more demanding, a generalisation of the approach formulated here should be implementable in 2d rather straightforwardly from a conceptual point of view.

**Flat space limit.** A problem that could be studied already with the 1d methods described in this paper is the extrapolation to large scaling dimensions and 2d flat-space scattering amplitudes. Reference [33] studied 1d bootstrap problems similar to those we encountered in Section 4.3. Through a two-fold extrapolation procedure, first extrapolating to infinite derivative order of the bootstrap, then extrapolating to large conformal weights, the authors were able to compute the flat space limit of the dual 2d S-matrix. In the companion paper [34] the result of this conformal bootstrap analysis was then related to extremal S-matrices determined from the S-matrix bootstrap. It would be very interesting to use the numerical bootstrap as a tool to learn more about multi-particle scattering amplitudes, especially since a multi-particle S-matrix bootstrap is so far missing (see however [35] for some steps towards a numerical S-matrix analysis of $n$-to-$m$-particle scattering amplitudes).

**Spectrum extraction.** The extraction of extremal spectra from the functionals obtained through SDP has been of vital importance since the early days of the numerical conformal bootstrap [36, 37]. It would be highly desirable to likewise systematically extract extremal spectra from the functionals of the six-point bootstrap by studying their kernels. In the framework of this paper, it should be noted that spectrum extraction is potentially more sensitive to the choice of $X$ matrices (see Section 3.2) than the bounds themselves. Indeed, while we heuristically observed that relaxations to SDPs with a reduced number of $X$ matrices can be applied without a strong impact on the bounds, the relaxation can lead to unphysical null-vectors of the extremal functionals, which annihilate only certain fixed descendant level $n$ blocks at a fixed exchange dimension $\Delta$.

**Analytic Functionals.** Beyond the numerical determination of extremal spectra, it would also be very interesting to get a better analytic understanding of extremal spectra and the

dual extremal functionals. Analytic functionals [38, 39] and Polyakov blocks [40, 41] for higher-point correlators, could provide valuable new perspectives for the ongoing effort to solve 1d crossing and $QFT_2/CFT_1$ [42].

## Acknowledgments

I thank António Antunes and Volker Schomerus for countless discussions, initial collaboration and comments on the draft of this manuscript. In particular, I would like to acknowledge the contributions of António Antunes to the perturbative computations performed in Section 4.4. Special thanks are also due to Connor Behan for his valuable and constructive feedback during the review process. Furthermore, I thank Julien Barrat, Carlos Bercini, Julius Julius, Alessio Miscioscia, Yu Nakayama, David Poland, Slava Rychkov and Petar Tadić for useful discussions. This project received funding from the German Research Foundation DFG under Germany's Excellence Strategy - EXC 2121 Quantum Universe - 390833306, the Collaborative Research Center - SFB 1624 "Higher structures, moduli spaces and integrability" - 506632645 and the Studienstiftung des Deutschen Volkes. I would like to thank the Yukawa Institute for Theoretical Physics and the Kavli Institute for the Physics and Mathematics of the Universe for hospitality during part of this work.

## A   Code

This appendix collects remarks on the code that is made available with this publication. The focus is on increasing usability of the code and not on a detailed description of its precise structure.

### A.1   `Generate_Functionals.nb`

The mathematica notebook `Generate_Functionals.nb` can be used to generate and store all derivative functionals used in this work as well as their decomposition according to the procedure described in Section 3. Upon running the notebook, it generates the file `Functionals_Λ2.txt` in the folder in which the notebook is stored. The file contains the functionals with up to 2 derivatives. The parameter $\Lambda$ can be changed in the first section of the notebook.

### A.2   `Generate_JSON_no1.py`

The python script `Generate_JSON_no1.py` generates the file `pmp.JSON` in the working folder, which serves as the input that has to be provided to `SDPB` in order to compute the bound discussed in Section 4.1. Several parameters need to be provided to `Generate_JSON_no1.py`. These are

- `fundir`: The folder in which the functionals generated by `Generate_Functionals.nb` (see App. A.1) are stored. In the example below, we assume that the functionals are stored in a folder "`functionals`" inside of the folder in which `Generate_JSON_no1.py` is called.

- `Lambda`: Derivative order $\Lambda$.

- `nTrunc`: Truncation in the $n$ parameter introduced through the decomposition described in Section 3.

- `gap`: Value of the gap to be tested.

- `h`: External scaling dimension $\Delta_\phi$.

- `prec`: Internal working precision.

- `extra_eval`: A string of integers separated by commas, that indicates which particular values of $n$ should be considered beyond $n \leq$ `nTrunc`. Entering $-1$ as part of `extra_eval` dictates that positivity should be imposed asymptotically for $n \to \infty$.

Using the `pmp.JSON` file generated by the example call

```
python3 generate_JSON_no1.py --fundir=functionals --Lambda=2 --nTrunc=20
--gap=0.187 --h=0.1 --prec=512 --extra_eval=100,-1
```

as input to `SDPB` should result in `SDPB` terminating with a dual feasible solution, indicating that for $\Delta_\phi = 0.1$ a gap of 0.187 can be excluded with $\Lambda = 2$ functionals. If on the other hand, the `gap` parameter is changed to 0.186, `SDPB` will not be able to find a dual feasible solution and e.g. terminate with the final status `maxComplementarity exceeded`. To improve the numerical bound and get closer to 0.18, higher derivative functionals need to be considered.

### A.3  `GFF_to_GFB_Flow.nb`

In section two of the notebook, the GFF spectrum and OPE data is computed by solving the truncated crossing equation for a scalar field $\phi$ of dimension $\Delta_\phi = 0.1$ with $\Lambda\text{max} = 10$ operators above the identity exchange. The parameters $\Delta_\phi$ and $\Lambda\text{max}$ can be adapted in the first lines of the second section.

After running the second section of the notebook, one can run the third section, which uses the GFF data that has been computed as initial data to flow to GFB. In the first two subsections i.e. 3.1 and 3.2, the flow is performed for all values of $\Lambda$ up to $\Lambda\text{MAX}$ and a folder called `h=0.1` (or `h=...` if $\Delta_\phi$ is changed) is created in which the resulting data is stored. Finally, an extrapolation to $\Lambda = \infty$ is performed. Running the notebook with the default parameters will take several hours.

In addition to $\Lambda\text{MAX}$, parameters $\Lambda 6$ and `dynamiclibPATH` are specified in the first lines of the third section of the notebook. The variable `dynamiclibPATH` needs to be set to the absolute path of the `arbitrary_precision_blocks.so` shared object that is described in Appendix A.4. $\Lambda 6$ on the other hand specifies which derivative order of the six-point bootstrap the notebook should generate the input data corresponding to the $\phi$ exchange in the central exchange of the six-point comb for. Upon running Section 3.3 of the notebook, six-point comb channel blocks and their derivatives are computed using `arbitrary_precision_blocks.so` and multiplied with the OPE data of the extremal flow.

The result of the computation is stored inside of the `h=0.1` folder in the form of txt files that serve as an input to the six-point bootstrap.

### A.4 `arbitrary_precision_blocks.c`

For the purpose of this publication, a short mathematica script that evaluates comb-channel blocks to a few digits precision by summing the series definition would have been completely sufficient. However, since we already had code available that efficiently evaluates the blocks to high precision, namely the `arbitrary_precision_blocks.c` of the ancillary files of this publication, we decided to use that code instead. The mathematica wrapper `arbitrary_precision_blocks.nb` can be used to call our `C` implementation of the blocks in mathematica. To do so, the user has to compile the `C` code into a shared object that has to be called `arbitrary_precision_blocks.so`.

### A.5 `Generate_JSON_GFF_to_GFB.py`

`Generate_JSON_GFF_to_GFB.py` is for the interpolation between GFF and GFB discussed in Section 4 what the script `Generate_JSON_no1.py` described in Appendix A.2 is for gap maximisation without identity. Hence, most of the parameters that need to be passed to it coincide with those discussed in Appendix A.2. Therefore, let us here only highlight one example call of `Generate_JSON_GFF_to_GFB.py` and then explain the three new parameters that occur in it.

```
python3 generate_JSON_GFF_to_GFB.py --h=0.1 --Lambda=2 --nTrunc=30 --prec=512
--share=input_data --triple=2.1 --D1=0.6 --extra_eval=100,-1
```

To sucessfully run the script with the arguments chosen above, the working folder must contain a folder called `input_data`. The `input_data` folder in turn must contain a subfolder called `functionals` and a subfolder called `objectives`. Inside of `input_data/funtionals`, one has to place the file `Functionals_Λ2.txt` generated by `Generate_Functionals.nb` (see App. A.1). Inside of `input_data/objectives`, the file `Λ_2_Δϕ_0.1_Δ1_0.6.txt` generated by `GFF_to_GFB_Flow.nb` (see App. A.3) has to be placed.

Finally, let us comment on the arguments `-triple` and `-D1`.

- `D1`: Dimension $\Delta_{\phi^2}$ of the lowest lying double twist operator (note that only values for `D1` can be used for which the necessary input data has been computed using `GFF_to_GFB_Flow.nb`).

- `triple`: Dimension $\Delta_{\phi^3}$ of the lowest lying double twist operator (in principle arbitrary positive real number).

With the input data generated by the example call of `Generate_JSON_GFF_to_GFB.py` given above, running `SDPB` should result in convergence to a primal-dual optimal solution putting an upper bound of 0.110 on the square of the four-point function $F_{\phi^3}$ of the four-point maximisation problem for $\Delta_\phi = 0.1$, $\Delta_{\phi^2} = 0.6$ and $\Delta_{\phi^3} = 2.1$.

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
