# Peer review of "Sparsity in the numerical six-point bootstrap"

_SciPost Physics_

## Round 1 · Referee Report · Anonymous (Referee 1) · 2025-8-26

Strengths

1-The author presents a clear motivation for this line of work, explains the newly developed method in detail, illustrates the method with a simple example, and studies its application to correlators that interpolate between the Generalised Free Fermion and the Generalised Free Boson in one dimension, where new bounds are obtained.

Report

In this paper the author develops a new method for bootstrapping six-point CFT correlators with semidefinite programming using sparse banded matrix decomposition. The paper certainly meets the criteria for publication, but there are a few very minor points that I think should be clarified.

Requested changes

1-In eq. (3.48), there is a parameter $n$ that counts the constraints used in SDP. It is not specified what the maximal value of $n$ is for the constraints used in the later SDP calculations. Also, similarly to the old six-point bootstrap, using finite $n$ truncates the set of constraints for fixed $\Lambda$ which could be stated explicitly. Is it clear whether going to higher $n$ would affect the bounds shown in the paper?

2-Could the author comment on how sensitive the results are to the choice of the
$X$-matrices given in eqs. (3.46) and (3.47)?

3-It is not entirely clear what the purpose is of the sections with steps 1, 2, and 3, where the author explains how to construct $X$-matrices that are not used later. Only the $X$-matrices given in step 4 are actually used in the calculation. Perhaps the first three steps could be moved to the appendix?

Recommendation

Ask for minor revision

  • validity: top
  • significance: high
  • originality: high
  • clarity: good
  • formatting: excellent
  • grammar: perfect

Author:  Sebastian-Philip Harris  on 2025-09-05  [id 5787]

(in reply to Report 1 on 2025-08-26)

  1. Thank you for pointing out that this was not addressed in the previous draft. I comment on the issue in a revised version of the paragraph "C matrices" of Section 3.2. This should also make the motivation for picking the specific conjugation matrices described in that paragraph clearer. I certainly regard this as an improvement of the text.

  2. Of course you could take any other basis of X matrices, i.e. work with appropriate linear combinations of the X matrices that I describe without changing anything. On the other hand, if you decide to not work with a full basis but rather restrict to a subset, this will generically influence your results. By first computing bounds using the full set of X matrices, you can develop some heuristics on which matrices are important and which can be omitted. I comment on this below eq. (3.48). Finally, there is the question of how many of the $X^{(n_0,I,J,d_\Delta)}$ matrices to include, which is brought up at the end of page 14. I added further comments on that in a new footnote on page 15, as well as in my reply to the Report of Referee 2.

  3. I rather prefer the current layout of Section 3.2. The referee is correct in his assessment that the X matrices constructed in step 1, 2 and 3 are not directly used. However, I consider them as necessary intermediate results for the construction of the final version presented in step 4. Of course one could instead just directly present the result of step 4 without any motivation and move step 1,2,3 into an appendix. But my impression is that the flow of the argument is more coherent in the way it is laid out right now. I will therefore not adapt the draft in this point. If the referee however has a strong opinion on this, I may of course reconsider my choice of presentation.

---

## Round 1 · Referee Report · Connor Behan (Referee 2) · 2025-9-3

Report

This paper investigates the consequences of crossing symmetry and unitarity for 6-point functions of primaries in 1d CFTs. Building on a previous paper by the author and three collaborators, it opens the door to a range of rigorous numerical bootstrap studies using semidefinite programming. Along with the analytic higher-point bootstrap (and possibly other numerical techniques like severe truncation and the fuzzy sphere), this gives access to CFT data which is much harder to extract using 4-point functions. Indeed, the decompositions reviewed by the author show that a single 6-point function contains information about infinitely many 4-point functions. This leads to a crossing equation involving matrices which are infinitely large.

The main achievement is an improved handling of these large matrices. When the feasibility of the 6-point bootstrap was first demonstrated, this was done by using larger and larger cutoffs and checking for stabilization of the eigenvalues. Only the last step exploited the fact that the matrices had a bandwidth of $\Lambda$ equal to the number of derivatives of the crossing equation. By contrast, the present paper trades each infinitely large matrix for a collection of $(\Lambda + 1) \times (\Lambda + 1)$ matrices in the first step instead of the last. Apart from making the final functional action easier to diagonalize, this leads to a much more flexible semidefinite program. In particular, it can be solved with SDPB which has been optimized for high-precision computations on clusters. Interestingly, the termination condition when a functional cannot be found appears to be different from the one that usually appears in the 4-point case.

As an application of this approach, the author shows that a very natural optimization problem is solved by a linear combination of $\Phi^4$ and $\Phi^6$ deformations of a free scalar in $AdS_2$. It had been known for some time that a free scalar with only $\Phi^4$ maximized $\left < \phi \phi \phi^2 \right >$. Since the 6-point bootstrap makes it possible to further maximize $\left < \phi \phi \phi \phi^3 \right >$, this allows one to move along an additional axis where the $\Phi^6$ interaction is needed.

Higher-point bootstrapping gives access to a funamentally new set of problems and this paper both formulates and implements an algorithm which makes it much more efficient. It will be great to have it published once a few minor issues are addressed.

Requested changes

  1. Page 3 states that the original 6-point bootstrap "had to resort" to discretization instead of polynomial approximation but I cannot find a more thorough explanation of this later on. It would be nice to add one.

  2. On page 11, it would be nice to relate things back to the notation of section 2.2. Perhaps by writing $M^\alpha(\Delta)$ in (3.26) and stating that each of these is equal to $M^{(a,b,c)}(\Delta_{\mathcal{O}})$ for a particular choice of $(a,b,c)$.

  3. When these infinite matrices are tiled by $(\Lambda + 1) \times (\Lambda + 1)$ submatrices labelled by $n$, it is clear that each will have the $\Delta$ degree given in (3.27). On the other hand, the $n$ degree seems less clear. Is there a simple explanation for why each $M_n(\Delta)$ should be obtainable by plugging $n$ into some master polynomial matrix $M(\Delta, n)$?

  4. In the SDP of (3.48), $n$ is an integer while $\Delta$ lives on a half-line. This seems to be the reason why the codes need to sample one but not the other (supplemented by asymptotic flags like $\texttt{extra_eval}=-1$). Are missing values of $n$ an obstacle to complete rigor in this study (similar to missing spins in $d>1$ 4-point bootstrap)? It would be nice to comment on this.

  5. Also in (3.48), it seems that a specific choice for the number of $X^{n_0, I, J, d_\Delta}$ matrices was made. Some motivation for this should be mentioned.

  6. Unlike (3.48), the SDP of (3.44) uses the $M_n(\Delta)$ notation. The $n$ dependence here is polynomial as well though so perhaps $M(\Delta, n)$ would be better.

  7. Since the number of $n$ values is being truncated (with a parameter like $\texttt{nTrunc}$), it seems that the number of $x_{n,I,J,d_\Delta}$ variables in (3.37) will still not be infinite in practice. Is there another way to see that it is still highly desirable to exploit the polynomial dependence on $n$?

  8. I am not sure why $n_0$ goes from $0$ to $\Lambda - 1$ in (3.48) while it appears to take very different values in (3.37).

  9. As a minor point, the first mention of SDPB in the paper is very matter-of-fact. Some readers might want to know more about what it is.

  10. Some apostrophes should be added. Namely "let use" -> "let's use" on page 4, "SDPBs" -> "SDPB's" on page 8, "Newtons" -> "Newton's" on page 18 and "bootstraps" -> "bootstrap's" on page 26.

  11. Finally, the paper is not consistent between "a SDP" / "an SDP" and "labeled" / "labelled".

Recommendation

Ask for minor revision

  • validity: -
  • significance: -
  • originality: -
  • clarity: -
  • formatting: -
  • grammar: -

Author:  Sebastian-Philip Harris  on 2025-09-05  [id 5788]

(in reply to Report 2 by Connor Behan on 2025-09-03)

  1. There are several answers to the question of why we had to resort to discretization in the first paper. The weakest reason would be that polynomial SDP is not directly implemented in SDPA as it is in SDPB. Of course this is no real obstacle, since we can write a small piece of code that chooses a basis of polynomials and translates polynomial SDPs into a problem that can directly be fed into SDPA. In fact, we did exactly that in the old paper in order to impose asymptotic conditions that are sufficiently strong to ensure convergence of the truncation scheme. Another stronger reason is that, without the matrix decompositions presented in the new paper, it is not clear to me how one would conjugate the matrices appearing in the SDP in such a way that you directly get polynomial entries with a low degree bounded by $\Lambda$ as in eq.~(3.27). I would therefore expect that in the old setup we would have been forced to consider polynomials of much higher degree, increasing the computational cost. This ties together with what I consider to be the strongest reason for why we had to use discretization: The matrix size was a limiting factor in the old implementation. To ensure convergence of the truncation scheme we had to work with matrices that had several hundreds of rows and columns. If we had decided to treat the Delta dependence not by discretization but by polynomial SDP this would have further increased the matrix size by roughly an order of magnitude. Anticipating this issue, we worked with Delta discretization. I hope that the reasons that I have just provided sufficiently answer your question. Personally, I do not think that there is anything interesting to learn for the future from what I just wrote. This is why I decided not to further expand on this issue in the draft. If after reading my answer you have a different opinion, I am of course open to reconsidering this choice.

  2. I agree. To address this point, I replaced the vague formulation "the relevant matrices" before eq.(3.26) by the more precise formulation "the matrices $M^{(a,b,c)}(\Delta)$ defining the sum rules (2.12)" and then replaced $M$ by $M^\alpha$ in the equation. Also, this made me realise that there was a inconsistency in the usage of square and curved brackets between section 2 and 3, which I now also fixed. Thank you for your comment.

  3. Actually, the explanation for the bound on the degree in n given in (3.27) is the same as the explanation for the bound on the degree in Delta. The (2-j-k) part just arises from the (2\Delta)_n n! in the projectors onto conformal multiplets. The extra \Lambda arises because each derivative just generates at most one power of Delta or n (see eq. (2.9)). Apart from this general argument, one can of course also check this polynomial dependence by simply looking at explicit expressions for the functionals that are computed in the Generate_Functionals.nb notebook.

  4. You are completely right to compare the discrete parameter n with the role that spin plays in the higher dimensional four-point bootstrap. In response to comment 1 of referee 1, I added some remarks on this in the paragraph "C matrices", which hopefully also address the point that you are making.

  5. That $X^{(n_0,I,J,d_\Delta)}$ matrices are included only for $n_0$ only up to $\Lambda-1$ is indeed a choice that I made and in principle one could include more of these matrices. I commented on this to some degree at the end of page 14. However, I agree that this was not sufficient. I now added a footnote with further comments on page 15.

  6. Thank you, this is a good observation. I corrected the notational inconsistency. As a side effect, the notation in (3.45) is now also less cluttered.

  7. I understand your question as: "Why is it desirable to use X matrices exploiting the polynomial dependence on n rather than truncating in n and then take X matrices of the type constructed in step 2". My answer to that would be two fold. First, for large enough nTrunc the procedure I chose leads to a smaller number of X matrices (which is desirable for numerical efficiency). Secondly, which I consider much more relevant, there is an important qualitative difference between the kind of SDP suggested by the question and the SDP used in my paper. As nTrunc is increased, both the number of parameters and the number of constraints grow in the former. In the latter, the parameters stay the same and only the number of constraints grow. Therefore it is easier to formulate a convergent algorithm (and also to check its convergence) in the latter case. This is the main reason why I chose to exploit the polynomial n dependence in the way described in the paper.

  8. In (3.37) the parameter $n_0$ runs over its full range. In (3.48) its range got restricted and I only pick a small subset of $X^{n_0,I,J,d_\Delta}$ matrices. The missing $X^{n_0,I,J,d_\Delta}(\Delta)$ are replaced by the $X^{(I,J,d_\Delta,d_n)}(\Delta,n)$ matrices. For further comments on this, see my answer to your remark 5.

  9. Good point. It now reads "On the other hand, \texttt{SDPB} -- the highly optimised solver designed specifically to address the needs of the bootstrap community -- is [...]".

  10. & 11. Thank you for noticing this. The spelling mistakes and grammatical errors that you pointed out are fixed now.

---

## Editorial Decision

resubmitted